# Restoration of high-sensitivity and adapting vision with a cone opsin

Michael H. Berry [1,2], Amy Holt[1], Autoosa Salari[1], Julia Veit[1,3], Meike Visel[1], Joshua Levitz[1,7], Krisha Aghi [3], Benjamin M. Gaub[3,8], Benjamin Sivyer[2,4], John G. Flannery [1,3,5] & Ehud Y. Isacoff[1,3,6]

Inherited and age-related retinal degenerative diseases cause progressive loss of rod and cone photoreceptors, leading to blindness, but spare downstream retinal neurons, which can be targeted for optogenetic therapy. However, optogenetic approaches have been limited by either low light sensitivity or slow kinetics, and lack adaptation to changes in ambient light, and not been shown to restore object vision. We find that the vertebrate medium wavelength cone opsin (MW-opsin) overcomes these limitations and supports vision in dim light. MW-opsin enables an otherwise blind retinitis pigmenotosa mouse to discriminate temporal and spatial light patterns displayed on a standard LCD computer tablet, displays adaption to changes in ambient light, and restores open-field novel object exploration under incidental room light. By contrast, rhodopsin, which is similar in sensitivity but slower in light response and has greater rundown, fails these tests. Thus, MW-opsin provides the speed, sensitivity and adaptation needed to restore patterned vision.

[1] Department of Molecular and Cell Biology, University of California, Berkeley, CA 94720, USA. [2] Department of Physiology and Pharmacology, Oregon Health and Sciences University, Portland, OR 97239, USA. [3] Helen Wills Neuroscience Institute, University of California, Berkeley, CA 94720, USA. [4] Department of Ophthalmology, Casey Eye Institute, Oregon Health and Science University, Portland, OR 97239, USA. [5] School of Optometry, University of California, Berkeley, CA 94720, USA. [6] Bioscience Division, Lawrence Berkeley National Laboratory, Berkeley, CA 94720, USA. [7] Present address: Department of Biochemistry, Weill Cornell Medicine, New York, NY 10024, USA. [8] Present address: Department of Biosystems Science Engineering, ETH Zürich, Mattenstrasse 26, Basel 8092, Switzerland. Correspondence and requests for materials should be addressed to E.Y.I. (email: ehud@berkeley.edu)

nherited retinal diseases (IRDs) lead to the progressive loss of rods and cones, beginning in the peripheral retina and underlie blindness in children and adults into middle age[1]. Age-related macular degeneration (AMD), the leading cause of severe vision loss in people over 50 years of age, also results from loss of photoreceptor cells, but in the central retina, impairing highest acuity vision[2]. IRDs can result from defects in >50 genes[3,4], and there are no clear single gene defects that cause AMD[5], making gene replacement costly or ineffective. Furthermore, gene replacement can slow progression but does not help in late stage disease.

An alternative approach is to endow light sensitivity to downstream neurons of the inner retina that survive following photoreceptor loss[6,7], either with synthetic photoswitches, which actuate native ion channels[8–12], or by genetic introduction of a light-sensitive signaling protein. A number of light-sensitive signaling proteins have been tested, including two microbial opsins, the ion channel channelrhodopsin and ion pump halorhodopsin[13–18], chemically engineered mammalian receptors[19,20] and two G-protein coupled receptor (GPCR) opsins that are native to the retina, rhodopsin of rod photoreceptor cells and melanopsin of intrinsically photosensitive retinal ganglion cells[21–25]. These light-gated systems, when delivered to the surviving neurons of the blind retina using adeno-associated viruses (AAVs), restore light sensitivity, transmission of light-driven activity to higher order visual centers in the brain, and both innate and learned visually guided behaviors. However, each of these approaches encounters a significant limitation. The microbial opsins and the chemically engineered receptors have fast kinetics (~50–200 ms) and follow high frequency modulation of light (~20 Hz) but have a low sensitivity to light (requiring intensities of very bright outdoor light). The sensitivity is so low that a clinical trial for channelrhodopsin requires intensifying goggles[26,27], risking retinal photo-toxicity[28,29]. In contrast, ectopic expression of rhodopsin and melanopsin are extremely sensitive to light (responding to indoor light), but so slow (seconds to tens of seconds)[21–24] that they may not support patterned vision given eye motion and movement of the subject and of visual objects. Moreover, the systems tested to date operate over a relatively narrow range of intensities. In contrast, normal photoreception combines speed with high sensitivity and adaptation that permits a sharp intensity-response curve to shift over 9 orders of magnitude. The sensitivity and adaptation emerge from the properties of G-protein signaling pathway of photoreceptor cells that both amplifies and modulates the light response. Our goal was to overcome the shortcomings of current vision restoration approaches by searching for a GPCR that could provide fast, sensitive and adapting light responses to surviving cells of the blind retina.

Cone opsins are G-protein coupled receptors of cone outer segments in the vertebrate retina. Like the rhodopsin of rods, cone opsins are highly sensitive and adapt to light over many orders of magnitude. While rods dominate the periphery of the retina and play a major role in vision under low light conditions, cones are densely arrayed in the fovea, where they mediate high acuity central vision, in both indoor and outdoor light.

We find that, when virally delivered to retinal ganglion cells (RGCs), medium wavelength cone opsin (MW-opsin) is as sensitive to light as rhodopsin under physiological stimulation parameters, but displays 10-fold faster kinetics. MW-opsin restores discrimination between flashing and constant light and between line patterns of different orientation, in both static and moving displays on a standard LCD computer screen. Rhodopsin under the same behavioral parameters performed no better than blind animals.

To date, adaptive goggles have been used to make an optogenetic prosthetic operate under natural conditions of changing ambient light levels. Strikingly, the light response of MW-opsin in RGCs itself adapts to ambient light, and does so over a considerable range, normalizing to background luminance. This "built-in" adaptation expands visual function across naturally encountered variations in lighting, from indoor to outdoor light.

Although optogenetic and chemical therapies for vision restoration for photoreceptor degenerative disease have been under development for more than a decade, none has yet been shown to restore object vision. We find that novel object exploration in an open field environment, under conditions that depend on vision, is restored by MW-opsin, indicating that restoration of object vision under natural light. Thus, MW-opsin provides a unique combination of speed, sensitivity, and luminance adaptation and restores key aspects of natural vision. MW-opsin therefore represents a promising new biological prosthetic for patients suffering from degenerative retinal disease.

## Results

**MW-opsin restores fast and sensitive light responses.** Recent studies have established that vertebrate rhodopsin, found in rod outer segments, may be used ectopically to control G-coupled signaling in cultured cells, RGCs and ON bipolar cells, but runs down with repeated stimulation and deactivates slowly[30,31,21,22], raising concern that it may not support vision of natural scenes because of movement of the observer and surrounding objects or saturation and possibly photo-bleaching of the chromophore under photopic lighting conditions. We wondered if another vertebrate opsin would have faster kinetics while maintaining similar light-sensitivity to rhodopsin. We turned to a mammalian cone opsin, MW-opsin, whose activation of the tetrameric GIRK1 (F137S) G protein-coupled inward-rectifier potassium channel in HEK293 cells we found to decay 8-fold more rapidly and recover more completely than rhodopsin (Fig. 1a–c).

We tested MW-opsin in the retina of the *rd1* mouse, which has a mutation in the PDE-6-β gene, resulting in progressive loss of rod and cone photoreceptor cells. MW-opsin under control of the human synapsin promoter (*hSyn-1*), with a yellow fluorescent (YFP) C-terminal tag for tracking expression, was packaged in *AAV2/2(4YF)* and injected intravitreally at postnatal day 45–60 (Fig. 1d, e). Retinas isolated 4–8 weeks later showed expression to be pan-retinal, with a transfection rate of 45 ± 19% (SD) and localized to the soma and dendritic layers of ON-RGCs and OFF-RGCs (Fig. 1f, g and Supplementary Fig. 1), consistent with previous studies[32] and similar to expression of rhodopsin under identical parameters (Supplementary Fig. 2). Retinas were mounted on a multi-electrode array (MEA), with the RGC layer in contact with the electrodes, to test for light-evoked activity. Due to complete photoreceptor degeneration in animals ≥3-months-old[33], no light-evoked response was detected in the retina of control *rd1* littermates (Fig. 1h), with the exception of a few RGCs which displayed slow responses characteristic of intrinsically photosensitive RGCs[34]. In contrast, retinas from MW-opsin *rd1* animals displayed robust light-evoked increases in firing rate, consisting of a large fast, transient component and a small (~30% in size) slow component (Fig. 2a, Supplementary Fig. 3). Responses across retina were normalized using the Light Response Index (LRI = (peak firing rate in the light—average firing rate in dark)/peak firing rate in the light + average firing rate in dark)) adopted from Tochitsky and colleagues[11] and our earlier work[20]. The light responses ran down with repeated bouts of light stimulation, as expected following removal of the retinal pigment epithelium, a source of 11-cis-retinal. The run down was reduced by the addition of 9-cis-retinal (a stable analog of 11-cis-retinal) to the recording solution (Fig. 1j).

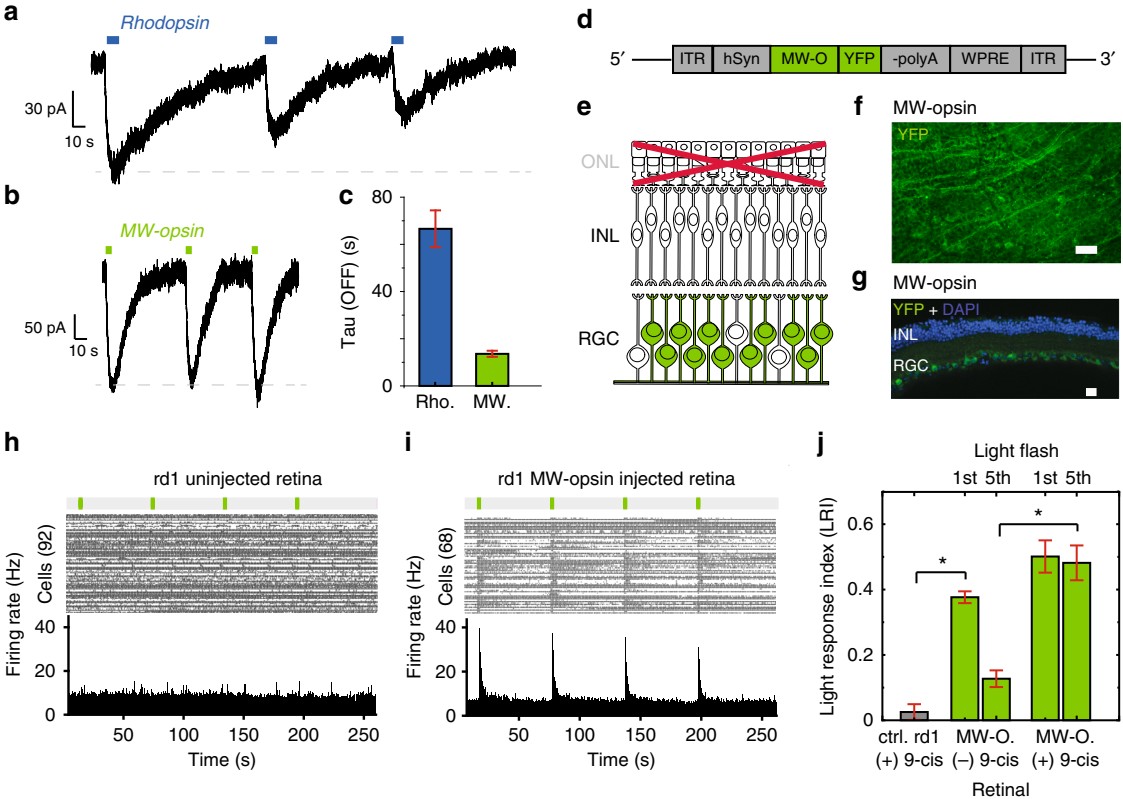

**Fig. 1** Expression and function of MW-opsin in HEK293 cells and RGCs of *rd1* mouse retina. **a**, **b** Representative traces of activation of homotetramer GIRK (F137S) channels by photo-stimulation of rhodopsin (**a**) or MW-opsin (**b**) measured in whole cell patch in 50 mM $[K^+]_{ext}$ at $V_H = -80$ mV in response to low intensity (1 mW cm$^{-2}$) pulses of light at 535 nm (for MW-opsin) or 500 nm (for rhodopsin). **c** Decay of photo-response (Tau OFF) for rhodopsin (blue) and MW-opsin (green). Values are mean + SEM; $n = 6$ (rhodopsin), 8 (MW-opsin) cells. **d** Viral DNA expression cassette. MW-opsin with YFP (green) under control of *hSyn-1* promoter. **e** Schematic of a degenerated *rd1* mouse retina with targeted RGCs highlighted (green). ONL outer nuclear layer, IPL inner plexiform layer. Photoreceptor degeneration denoted in light gray and red cross. **f**, **g** *En face* view of flat mount (**f**) and transverse slice (**g**) confocal images of MW-opsin expression of *rd1* mouse retina 4 weeks after intravitreal injection of *AAV2/2-hSyn-MW-opsin-YFP*. Images of YFP fused to C-terminal end of MW-opsin (green) show pan-retinal distribution (**f**) in RGC layer in relation to DAPI staining of nuclei (**d**, blue). Scales 60 μm (**f**) and 20 μm (**g**). **h**, **i** MEA recordings from representative uninjected control (**e**) and MW-opsin expressing (**f**) *rd1* mouse retina. (Top) Raster plot with spikes for each RGC (**e**: $n = 92$ cells; **g**: $n = 68$ cells). (Bottom) Peristimulus time histogram (PSTH). Light stimulation protocol: 4 pulses of light of 100 ms duration ($\lambda = 535$ nm, enlarged green bars) separated by 60 s dark intervals. **j** Normalized Light response Index (LRI) for *rd1* retina without (gray) and with MW-opsin expression (green) (gray: $N = 6$ retinas, $n = 295$ cells; green: $N = 8$ retinas, $n = 323$ cells). LRI for 1st and 5th light flash without ($N = 3$ retinas, $n = 106$ cells) and with ($N = 6$ retinas, $n = 257$ cells) 9-cis retinal. Light intensity 2 mW cm$^{-2}$. Wavelength: $\lambda = 535$ nm (MW-opsin), Values are mean + SEM. Cells are sorted units. Statistical significance assessed using Mann–Whitney U test (*$p \leq 0.01$)

MW-opsin *rd1* mouse retinas were highly sensitive to light, to a degree similar to rhodopsin (Fig. 2b, c), in the range of indoor light intensity, and ~1000-fold higher than channelrhodopsin[13,18] and halorhodopsin[16,17] (Supplementary Fig. 4). While similar in sensitivity to rhodopsin (Supplementary Fig. 5a) MW-opsin ran down less in response to repeated flashes (Supplementary Fig. 5b) and had faster kinetics: ~5-fold faster rise, ~3-fold shorter time to peak and ~7-fold faster decay following a light pulse (Fig. 2d, e). The time constant of rise, the time to peak, and the time constant of decay changed modestly with decreasing light intensities, maintaining the advantage in speed over most of the intensity range (Supplementary Fig. 5c–e)[21,22]. The rise and decay kinetics of the response in RGCs of *rd1* retina expressing MW-opsin resembled those of the RGC transient ON-response seen in *wt* retina, except that the former had a longer latency (Fig. 2e and Supplementary Fig. 5f, g). The fast response kinetics and sensitivity of MW-opsin suggested that it would respond to brief flashes of light. Indeed, illumination pulses as short as 25 ms, triggered responses that were ~50% of the maximal peak response (Fig. 2f, g), similar to what is seen in *wt* retina[35].

We examined contrast sensitivity in the excised retina and in primary visual cortex in vivo by measuring responses to full-field gray scale steps. In the excised *rd1* retina expressing MW-opsin, RGC activity changed in response to changes in brightness of as little as 25% (Supplementary Fig. 6a, b), approaching but not equivalent to the contrast sensitivity of the wild type retina (Supplementary Fig. 6c). In complementary in vivo experiments on *rd1* animals expressing MW-opsin in RGCs, we measured single unit responses and visually evoked potentials across the layers of primary visual cortex in awake, free running animals (Supplementary Fig. 7) and observed similar contrast sensitivity using a standard computer monitor (Supplementary Fig. 8). The cortical responses followed flash frequencies up to at least 4 Hz (Supplementary Fig. 9). The sensitivity and kinetics of the light responses imparted by MW-opsin in RGCs suggested that it may support visually guided behavior.

**MW-opsin restores innate light avoidance.** Sighted mice innately avoid illuminated areas, a survival mechanism to evade capture[36] that is lost following photoreceptor degeneration in *rd1*

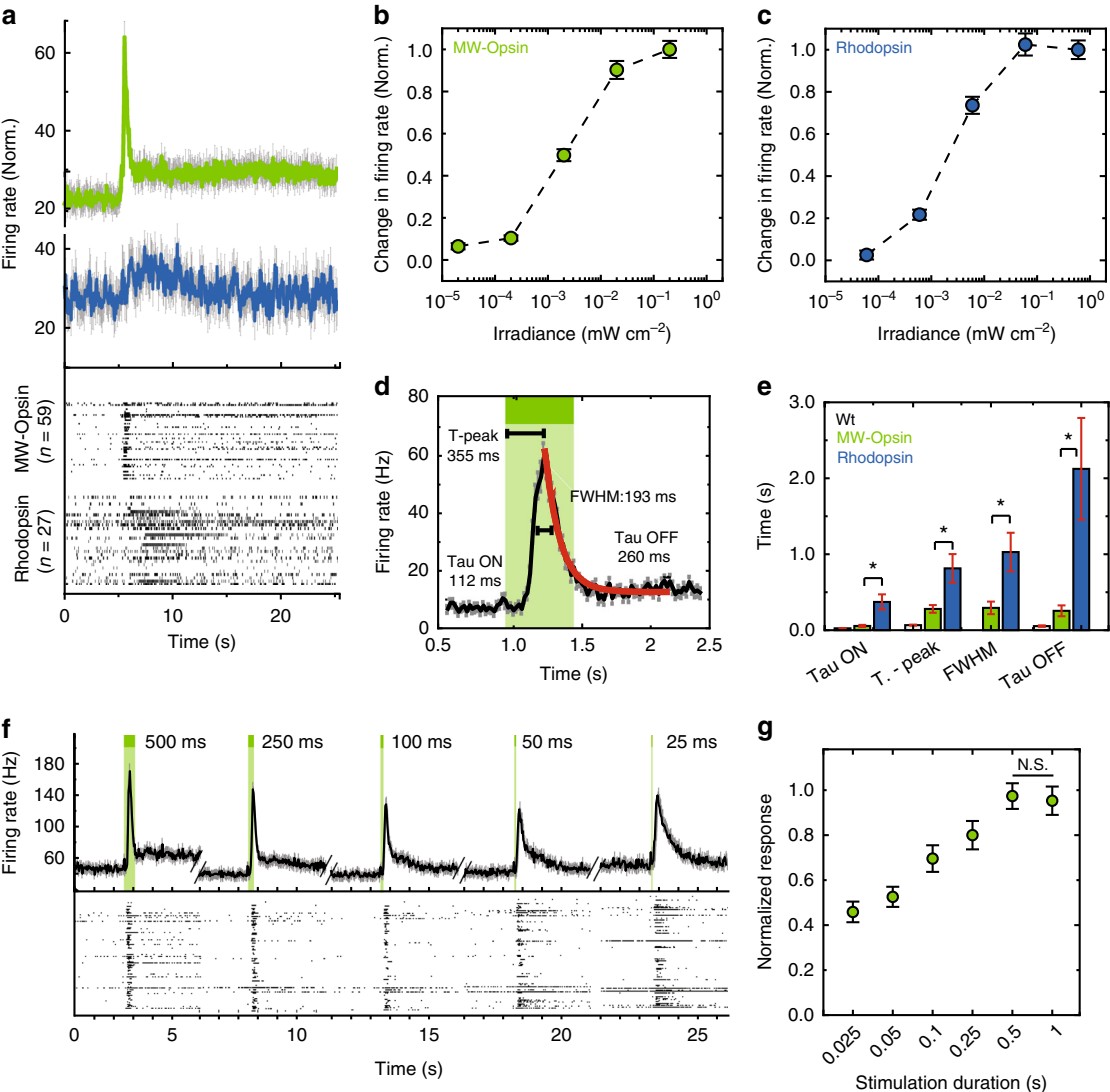

**Fig. 2** Light response in isolated *rd1* mouse retina with MW-Opsin in RGCs. **a** (Top) Average response to light flash of RGC population expressing MW-opsin (green) or rhodopsin (blue) in *rd1* mouse retina. (Bottom) Raster plot of average response of *rd1* mouse retina RGCs to 5 flashes of 100 ms duration light at 535 nm for MW-opsin ($n = 59$ cells) and 510 nm for rhodopsin ($n = 27$ cells) expressing in RGCs. **b, c** Light sensitivity for MW-opsin ($N = 6$ retinas) and rhodopsin ($N = 4$ retinas) in RGCs of *rd1* mouse retina. Peak firing rate normalized to maximum at highest intensity. **d, e** Time-course of light response. Population average traces with time from light onset to max excitation (time to peak: $355 + 21$ ms), exponential fits for excitatory phase (Tau ON: $112 + 25$ ms) and decay (Tau OFF: $260 + 31$ ms) and full width at half max (FWHM: $183 + 85$ ms) (**e**) for MW-opsin (**d**; **e**, green; $N = 5$ retinas, $n^c = 104$ channels), rhodopsin (**e**, blue; $N = 7$ retinas, $n^c = 134$ channels) and wt (**e**, white; $N = 3$ retina, $n^c = 97$ channels). **f, g** Dependence of MW-opsin light response on flash duration. **f** Representative retina light response ($n = 117$ cells): population average firing rate (top) and raster plot of unit responses (bottom). **g** Normalized peak responses for different stimulation durations ($2 \times 10^0$ mW cm$^{-2}$, $N = 2$ retinas, $n^c = 63$ channels). Light intensity $2 \times 10^{-1}$ mW cm$^{-2}$ unless specified, Wavelength: $\lambda = 535$ nm (MW-opsin) or 510 nm (rhodopsin). $N$ = number of retinas, $n^c$ = # of channels, $n$ = number of cells/units. Cells are sorted units. Values are mean + SEM. Statistical significance assessed using Mann–Whitney $U$ test (*$p \leq 0.05$)

mice[14,23]. To determine if this behavior could be restored, *rd1* mice expressing MW-opsin were tested in a behavior box consisting of adjoining light and dark compartments (Fig. 3a). The fraction of time spent in each compartment was recorded and compared to untreated *rd1* and *wt* mice (Fig. 3b, Supplementary Fig. 10a–c, Supplementary Table 1). The light compartment was illuminated with low intensity white light, equivalent to indoor office lighting (100 µW cm$^{-2}$). Untreated *rd1* mice spent ~45% of the time in the dark compartment, reflecting a slight location bias in favor of the release compartment (see Supplementary Methods) (Fig. 3b). In contrast, *rd1* mice expressing either rhodopsin or MW-opsin showed a strong preference for the dark compartment (~70%), similar to normally sighted *wt* animals (80%) (Fig. 3b and Supplementary Fig. 10a). When white light was

replaced with blue ($460 \pm 22$ nm) or green ($535 \pm 25$ nm) light and the intensity that was reduced to a lower end of the isolated retina intensity-response curves for MW-opsin and rhodopsin ($1$ µW cm$^{-2}$; Fig. 2b) both MW-opsin and rhodopsin expressing animals showed green light avoidance (Fig. 3c, left and Supplementary Fig. 10b), but only rhodopsin animals showed blue light-avoidance (Fig. 3c, right and Supplementary Fig. 10c), consistent with their absorption spectra[37].

**MW-opsin supports temporal light pattern discrimination.** We used a visually cued fear-conditioning paradigm to test the ability of animals to differentiate flashing from constant light. We used a single compartment behavioral apparatus with a low intensity

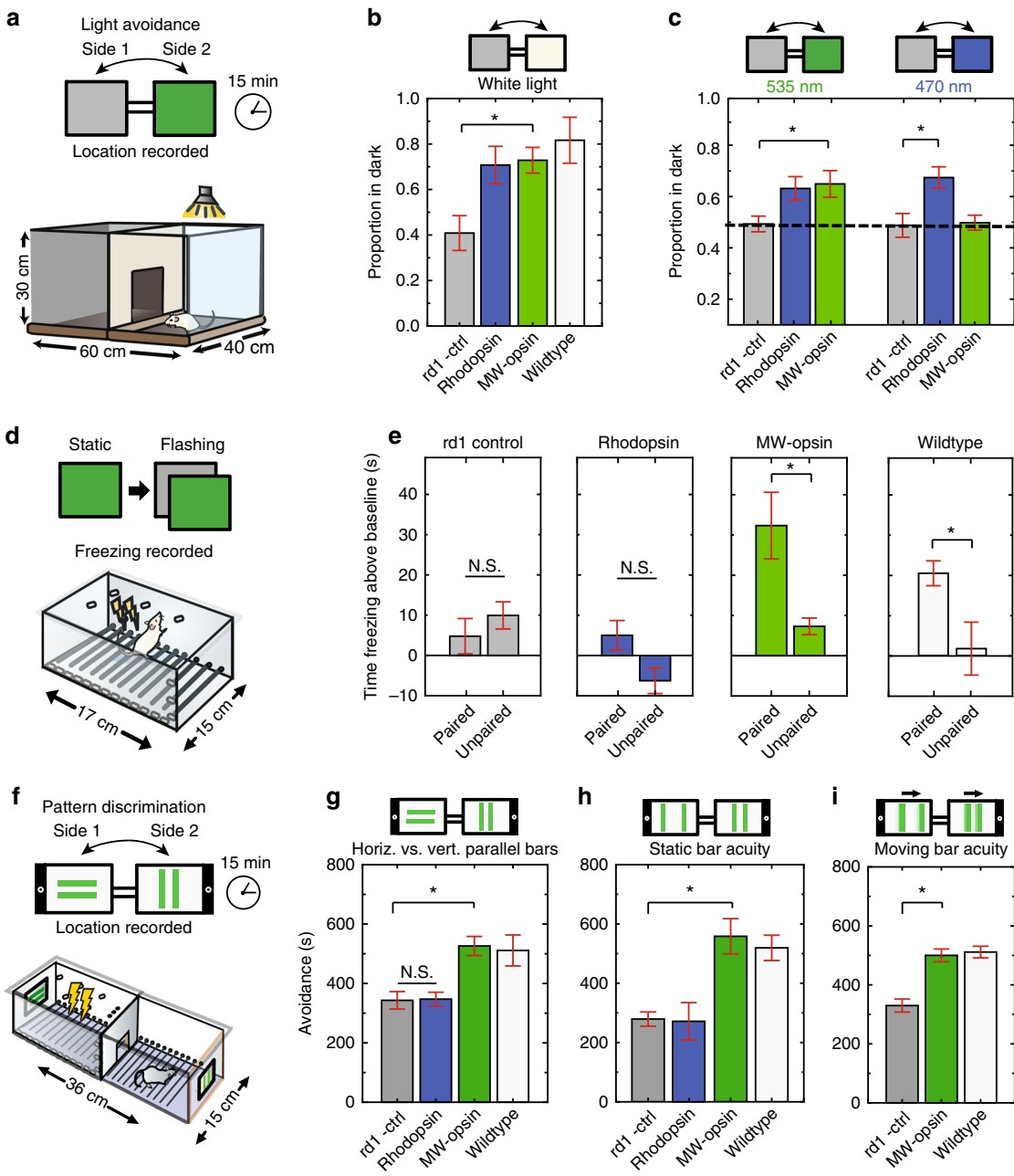

**Fig. 3** Light avoidance and learned visually guided behavior in *rd1* mice expressing MW-opsin or rhodopsin in RGCs. **a** Schematic of light/dark box for light avoidance test. **b**, **c** Respectively for **b**, **c**-left and **c**-right, proportion of time spent in the dark compartment (proportion of avoidance) for *rd1* control (gray; *n* = 5,8,5 mice), *rd1* expressing rhodopsin in RGCs (blue; *n* = 6,8,5 mice) or MW-opsin (green; *n* = 11,10,4), and *wt* mice (white; *n* = 5 mice) when illuminated with either (**b**) white light (100 µW cm⁻²), (**c**) 1 µW cm⁻² blue light (470 nm) (right) or green light (535 nm) (left). **d** Schematic of freezing response fear conditioning experiment. **e** Quantification of fear response for discrimination of temporally patterned stimulation. Time freezing above baseline is shown for when illumination transitions from static to 2 Hz frequency stimulation (100 µW cm⁻²) was paired or unpaired with a electric shock for control *rd1*, rhodopsin, MW-opsin, and *wt* mice (*n* = 4,6,12,10 paired, *n* = 7,8,8,8 unpaired). **f** Schematic of pattern discrimination experiment. Mice habituated at day 1, then exposed to electric shock in association with specific pattern of light projected to tablets and paired randomly in either chamber (conditioning days 2 and 3). On day 4 recall tested (time spent in each chamber), in absence of shock with light patterns reversed to avoid location bias (See Supplementary Fig. 11d). **g**–**i** Learned pattern discrimination. Time spent avoiding pattern paired with shock. **g** Horizontal vs. vertical parallel bars. Discrimination of parallel static (**h**) or moving (**i**) bars at distances of 1 vs. 6 cm. Respectively for **g**, **h** and **i**: *rd1* control (*n* = 8,5,16), *rd1* rhodopsin (blue; *n* = 8,6 mice), *rd1* MW-opsin (*n* = 17,11,6) and *wt* (*n* = 5,6,9). (25 µW cm⁻²). (Note, proportion of success for these experiments shown in Supplementary Fig. 11). Light intensity = 25–100 µW cm⁻²; Wavelength: = 535 nm (MW-opsin), 510 nm (rhodopsin) or white light (MW-opsin). *n* = number of mice. Values are mean + SEM. Statistical significance assessed using Student's two-tailed *t*-test with Bonferroni correction: *$p < 0.05$

(100 μW cm$^{-2}$) LED light that switched between constant and flashing (2 Hz) light. For each animal, either constant or flashing light was paired with a mild foot shock for 2 days (Fig. 3d) and freezing time was measured on day 3 in response to light cues in absence of foot shock[11,22,38]. Freezing time in this "paired group" was compared to that of an "unpaired group," in which training shocks were randomized (i.e., not paired consistently with a visual cue). Freezing times in untreated *rd1* mice did not differ between paired and unpaired conditions, consistent with the inability to see the visual cues (Fig. 3e, gray). In contrast, *rd1* mice expressing MW-opsin froze more in the paired condition, as observed in *wt* animals (Fig. 3e, green and white). Strikingly, *rd1* mice expressing rhodopsin did not differ between paired and unpaired conditions (Fig. 3e, blue). This suggests that, unlike blind mice expressing MW-cone opsin, rhodopsin mice cannot discriminate light flashing at 2-Hz from constant light, consistent with the slow light response kinetics observed in MEA (Fig. 2e).

**MW-opsin restores spatial pattern discrimination**. We asked if MW-opsin in RGCs would enable *rd1* mice to detect spatial light patterns. We used a behavioral chamber with two adjoining compartments (Fig. 3f), each with a low-intensity LCD tablet (iPad) mounted on a wall that displayed a pair of parallel lines: in one, the lines were oriented vertically (||) and in the other horizontally (=). For MW-opsin the wavelength was centered at 535 nm (520–560) and for rhodopsin at 497 nm (480–520). On day 1, mice were habituated to the compartments with the visual displays off. During a 2-day training period, an aversive foot shock was paired with either the vertical or horizontal lines (Supplementary Fig. 10d). On day 4 the locations of the stimuli were switched to avoid location bias and conditioned avoidance was tested. We found that *rd1* animals expressing MW-opsin showed avoidance of the aversive visual cue at a level significantly higher than did untreated *rd1* controls, and similar to *wt* mice (Fig. 3g and Supplementary Fig. 10e). *Rd1* animals expressing rhodopsin behaved like the blind untreated *rd1* controls. These results indicate that MW-opsin restores the ability to recognize spatial light patterns, but rhodopsin does not.

We next asked if mice could discriminate differences between lines of identical orientation but different spacing, a visual task adopted from tests of visual acuity in humans and animals[39,40]. Parallel vertical lines were separated by distances of 1 or 6 cm. As above, an aversive foot shock was paired with one of the stimuli during the training period on days 2 and 3, and recall was tested on day 4. We found that *rd1* mice expressing MW-opsin are able to distinguish between the two patterns with a performance that is similar to that of *wt* mice, whereas rhodopsin expressing animals are similar to untreated *rd1* mice (Fig. 3h. Supplementary Figs. 10f and 12). MW-opsin also supported line differentiation when the parallel lines were in motion (1 cm/s) (Fig. 3i and Supplementary Fig. 10g).

**Light adaptation of MW-opsin response and visual function**. A fundamental characteristic of vision is the ability to distinguish objects across a wide range of ambient light intensities[41,42]. We wondered whether some aspect of adaptation would operate in the *rd1* retina expressing MW-opsin. Isolated retinas were kept in complete darkness for 15 min (dark-adapted) and then tested in a series of brief (100 ms) flashes of green light (535 + 25 nm) at 60-s intervals and over a range of intensities. Retinas were then adapted for 5 min to a moderate indoor light level (light-adapted; white light at 100 μW cm$^{-2}$) and retested. We first examined the kinetics of the light responses. The light response decayed rapidly, as shown above, displaying similar response kinetics for both the light and dark-adapted retina (Fig. 4a Supplementary Fig. 12a, b).

The intensity-response curve showed a high light sensitivity in the dark-adapted retina (responding at ~0.5 μW cm$^{-2}$) and lower sensitivity after adaptation to moderate light (responding to ~200 μW cm$^{-2}$) (Fig. 4b, c and Supplementary Fig. 12a, b). This adaptation represented a shift by 2–3 orders of magnitude on the intensity axis (321 + 89, N = 3) (Fig. 4b, c).

We asked if MW-opsin would provide visually useful light adaptation in the behaving animal, first in the context of light avoidance behavior. *Rd1* mice expressing MW-opsin were either dark adapted or light adapted to bright indoor illumination for 1 h (white light, 1 mW cm$^{-2}$/535 nm light component, 50 μW cm$^{-2}$) (Fig. 4d). They were then tested immediately in the two-chamber light-dark box for light avoidance at either 1 μW cm$^{-2}$ (dim) or 100 μW cm$^{-2}$ (bright). The light adapted MW-opsin expressing *rd1* mice showed stronger light avoidance with the brighter test light, whereas the dim test light produced a high level of light avoidance in the dark-adapted animals (Fig. 4e and Supplementary Fig. 12c). Pattern recognition was also influenced by light adaptation. *Rd1* mice expressing MW-opsin were trained by pairing mild foot shock with a display of parallel lines at one of two spacing, similar to that described above (Fig. 3h and Supplementary Fig. 10d). They were then dark-adapted (1 h) or light-adapted (1, 4 or 8 h) before testing. We found that the dark-adapted animals were able to discriminate between the line patterns whether they were presented at the low (0.25 μW cm$^{-2}$) or moderate (10 μW cm$^{-2}$) indoor intensity (Fig. 4f and Supplementary Fig. 12d), but that light-adapted animals only succeeded with the brighter test line patterns and were identical in performance between the groups that were light adapted for 1, 4, and 8 h (Fig. 4f and Supplementary Fig. 12e). The results show that spatial pattern recognition mediated by MW-opsin is adaptive over a range of natural light intensities.

**MW-opsin restores novel object exploration**. Our experiments above show that MW-opsin enables pattern recognition across a wide range of light intensities using illuminated displays. We wondered how it would operate in a natural environment, where ambient, incidental light illuminates three-dimensional objects. To address this, we employed an open field arena that is commonly used to test novel object recognition and exploratory behavior[43,44]. Mice naturally avoid open spaces and maintain proximity to walls of their environment. Exploratory excursions from these places of safety can be motivated by novel stimuli. Although mice employ multiple sensory modalities during exploration, vision has been shown to be critical for spatial navigation[45]. Our arena consisted of a cube containing two distinct novel objects. The mouse was placed against the arena wall, far enough from the objects, which themselves were far enough apart, so that the chance of an accidental encounter was low whether the animal walked along the wall or explored the other object. We filmed *rd1* untreated, *rd1*-sham injected, *rd1* expressing rhodopsin or *rd1* expressing MW-opsin mice, as well as *wt* animals. Their movements were tracked for 10 min the first time that they were placed into the arena (Fig. 5a–d). We found that *wt* animals travel 1.6-fold farther and moved at an average velocity 1.59-fold faster than blind *rd1* animals, consistent with the known visual component of exploratory behavior. Strikingly, like *wt* animals, *rd1* animals expressing MW-opsin traveled farther (by 1.42-fold) and faster (by 1.41-fold) than their untreated *rd1* littermates (Fig. 5e, f), suggesting that MW-opsin supports normal novel object exploration. To analyze this further, we focused on aspects of exploratory behavior that most likely depend on vision at a distance; the latency to exploration of the novel objects and the velocity and distance traveled on the excursions to the objects. Sham injected and rhodopsin expressing *rd1* mice performed

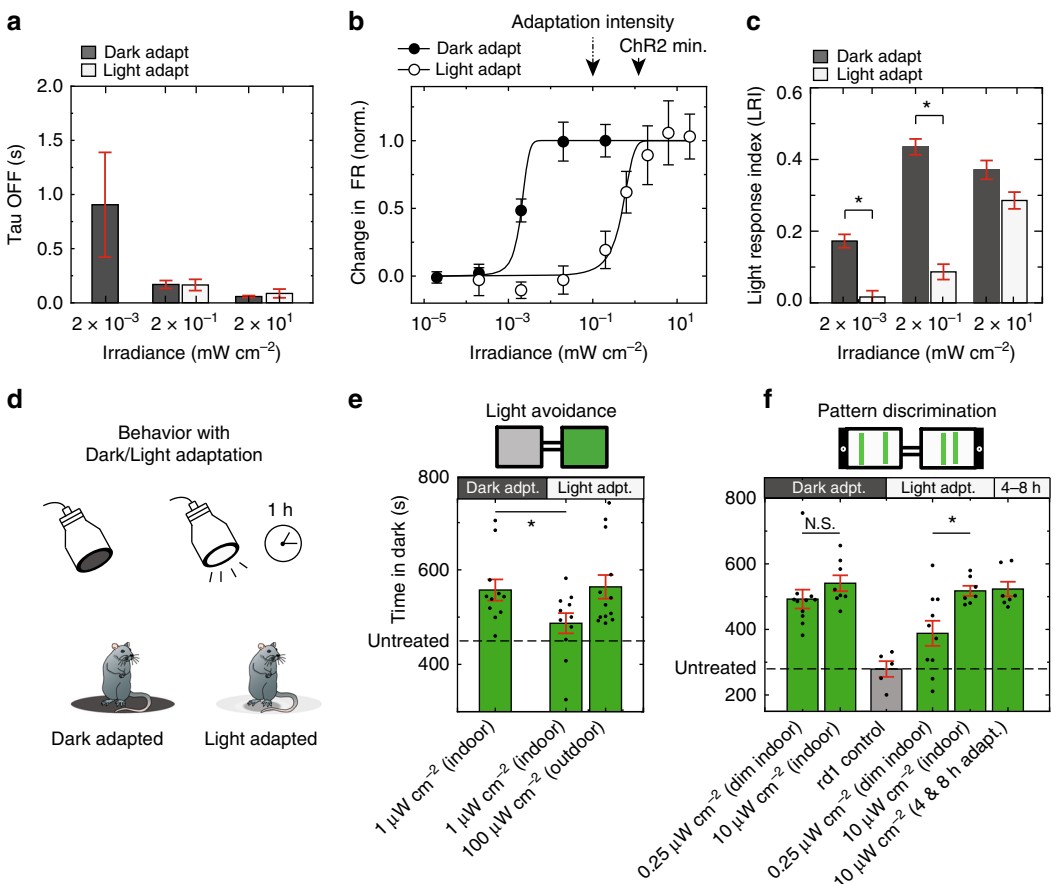

**Fig. 4** Light adaptation in RGC activity and visually guided behavior by MW-opsin. **a–c** MEA recordings in isolated retina of RGC light response mediated by MW-opsin in RGCs of *rd1* mouse retina show sensitivity difference with retina adapted to dark *versus* light. **a** Light response decay (Tau OFF) as a function of flash intensity in dark versus light adapted condition (*N* = 3 retinas, *n*^c = 88 channels). **b** Example intensity-response curve for representative retina first dark adapted (filled symbols) then light adapted (open symbols) (*n* = 56 cells). White light adaptation. ChR2 minimum value from Bi et al.[13] and Sengupta et al.[18]. **c** Average (error bars are SEM) normalized Light response Index (LRI) at 3 flash intensities in same retina, first dark adapted and then light adapted (*N* = 3 retinas, *n*^c = 88 channels). **d–f** Behavior shows light adaptation in visually guided tasks. **d** Schematic of adaptation to dark or light prior to testing of innate avoidance behavior or learned pattern discrimination behavior. **e** Proportion of time spent in the dark compartment (proportion of avoidance) under 100 μW cm⁻² (bright light) or 1 μW cm⁻² (dim indoor light) following 1 h. of adaptation to dark (*N* = 11 mice) or adaptation to light (white light; 1 mW cm⁻²/535 nm spectral component; ~50 μW cm⁻²; *N* = 11,13 mice). **f** Learned pattern discrimination of parallel bars spaced at distances of 1 versus 6 cm displayed at low (0.25 μW cm⁻²) or indoor (10 μW cm⁻²) light levels following 1 h. of adaptation to dark (*N* = 11 and 8 mice for each display) or light (white light; 1 mW cm⁻²/535 nm spectral component; 50 μW cm⁻²; *N* = 10 and 7 mice for each display). Performance was also reported in cohorts experiencing 4 and 8 h. of light adaptation (*N* = 7 mice). Dotted line denotes average performance of untreated *rd1* control mice and performance (gray *N* = 5 mice) reproduced from Fig. 3h for reference and comparison. Wavelength: *λ* = 535 nm. *N* = # of animals, *n* = # of retina, *n*^c = number of channels. Cells are identified as sorted units. Values are mean + SEM. Statistical significance assessed using Mann-Whitney *U* test (*$p < 0.01$). Student's two-tailed *t*-test with Bonferroni correction: *$p < 0.05$

similarly to untreated *rd1* animals, but MW-opsin mice reached the first and second objects in 5.12-fold and 4.25-fold shorter times, respectively (Fig. 5g, h), moved at velocities that were 2.2-fold and 1.89-fold faster to the first and second objects, respectively (Fig. 5i, j), and took shorter pathways that were 0.60-fold and 0.55-fold the distance to the first and second objects, respectively (Fig. 5k, l), as compared to untreated *rd1* mice. In each of these measures, MW-opsin expressing *rd1* mice reached levels that were similar to those of *wt* animals (Fig. 5e–l). These results suggest that MW-opsin in RGCs provides previously blind animals with naturalistic vision of objects under ambient light.

## Discussion
Until now, optogenetic tools for vision restoration have had one of two significant shortcomings. The microbial opsins, chan-nelrhodopsin and halorhodopsin[15–18,46], and the light-engineered

mammalian receptors[20,47] respond rapidly to light (in milli-seconds) and so should support "refresh rates" of sufficient speed for vision in motion, but they require such intense light as to risk damage to the retina. Conversely, the opsins from rods and intrinsically photosensitive RGCs—rhodopsin and melanopsin—are sufficiently sensitive to enable function in dim room light, but are very slow (hundreds of milliseconds to tens of seconds)[21–24]; slow enough to raise the concern that patterned vision may not be possible, as indeed we show here in tests of visual pattern recognition.

We find that MW-opsin expressed in RGCs of blind *rd1* mice overcome these shortcomings, providing the combined speed and sensitivity to enable both static and moving pattern recognition in dim light. We also find that a key property of practical vision, light adaptation, is provided by MW-opsin. This adaptation covers 2–3 orders of magnitude, from dim room light to outdoor light. Finally, MW-opsin does more than support the recognition

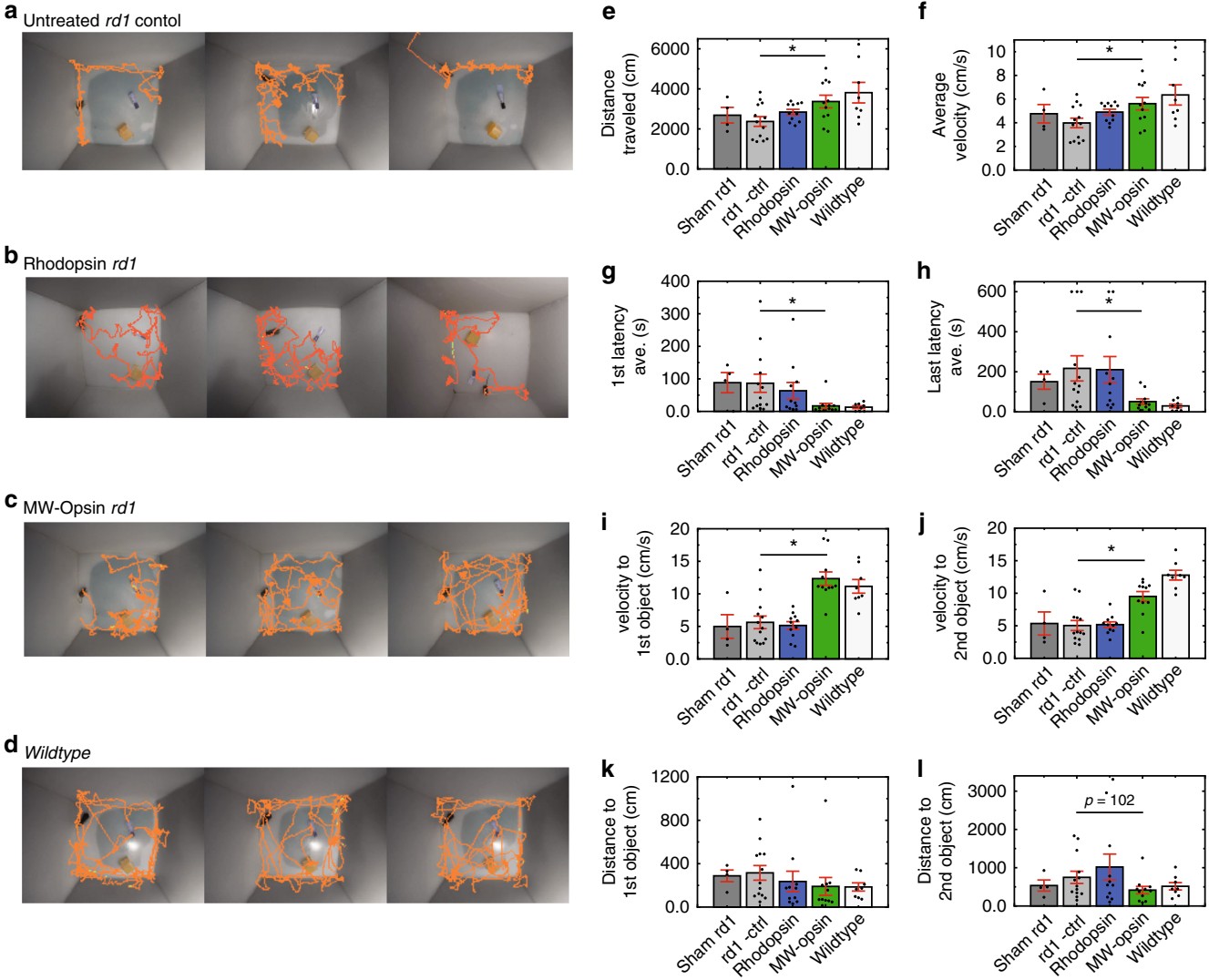

**Fig. 5** Restoration of visually guided exploratory behavior by MW-opsin. **a–d** Open field behavioral arena containing two novel objects with traces of the first minute locomotion track from 3 representative animals per condition: untreated *rd1* mice (**a**), rhodopsin expressing *rd1* mice (**b**), MW-opsin expressing *rd1* mice (**c**), and *wt* mice (**d**). **e**, **f** Total distance traveled and average velocity of mice during 10 min of exploration in the novel object behavior box. **g–l** Latency to exploration of first object (**g**), cumulative latency to exploration of the second object (**h**), velocity of travel to the first (**i**) and second object (**j**), distance traveled to first (**k**) and second object (**l**) for *rd1*-sham injected (dark gray) ($n = 4$), *rd1* untreated (gray) ($n = 13$), rhodopsin (blue) ($n = 11$), MW-opsin (green) ($n = 11$) and *wt* (white) ($n = 8$). Values are mean + SEM. Statistical significance assessed using a two-tailed $t$ test (*$p < 0.05$)

of abstract line patterns displayed on an LCD computer screen, it restores visually guided exploration of novel objects under normal incidental room light. The combination of sensitivity, speed and adaptation, the reduced risk of immune reaction due to use of a native retinal protein, and the restoration of patterned vision make MW-opsin unique among methods for restoring vision.

Cone opsins are found in the cone photoreceptor cells of the retina and are responsible for color vision under photopic light conditions. The cone photoreceptors dominate our central (foveal) vision and are responsible for fine visual acuity of the fixating eye. There are three classes of cone opsins in humans: short-wavelength, medium-wavelength, and long-wavelength, corresponding to different absorption maxima and covering different parts of the visible spectrum. Like rhodopsin found in rod photoreceptors, the cone opsins are GPCRs that use isomerization of the chromophore 11-*cis*-retinal to induce conformational changes leading to activation of a second messenger cascade that results in the gating of downstream effector ion channels.

In their native photoreceptor cell environment, the cone opsins differ from rhodopsin in a number of ways: they regenerate more rapidly from opsin and 11-cis retinal following bleaching, have faster thermal isomerization, and are faster in the formation and decay of functional pigment intermediates such as meta II[48–52]. The question is whether these differences reflect an environment unique to rods and cones and if the differences would remain if the opsin were moved to another cell type in the retina, where its G protein, regulatory systems, effectors and other specialized subcellular signaling complex, may be unfamiliar or absent. As shown earlier for short and long wave cone opsins[31], we find that MW-opsin expressed in HEK cells activate GIRK channels more rapidly than rhodopsin, suggesting that the higher speed reflects an intrinsic property of the cone opsin. But would this translate into more rapid signaling in RGCs? When MW-opsin and rhodopsin are individually expressed in *rd1* mouse retina under identical conditions, both the on and off kinetics of excitation are faster in MW-opsin, with a combined effect of a ~10-fold faster

signaling speed. However, the MW-opsin response of RGCs is delayed compared the natural ON-response of *wt* retina. Nevertheless, at moderate light intensity, MW-opsin responds to a flash of light as brief as 25 ms. We conjecture that the higher speed enabled *rd1* mice expressing MW-opsin to distinguish flashing from steady light, as well as *wt* animals, while *rd1* mice expressing the slower rhodopsin were unsuccessful. Modifications that reduce the delay to the response may further improve performance.

We find that MW-opsin expressed in RGCs is highly light sensitive, similar to the other opsins of the mammalian retina, rhodopsin, and melanopsin[21–24] and are able to operate in normal room light. In contrast, microbial channelrhodopsin and halorhodopsin are 1000–10,000-fold less sensitive[16,17]. This insensitivity can be overcome by intensifying goggles, however, this brings the retinal illumination very close to the safety threshold determined by the European Commission guidelines for limited exposure of artificial optical radiation in patients[53]. Improved microbial opsins have been shown to have enhanced sensitivity[18,54] and may reduce this problem in the retina, but do not appear to increase sensitivity to the level of intrinsic photoreceptors like that achieved by the ectopic expression of vertebrate opsins of the retina.

Prior purely optogenetic approaches have been shown to restore light-evoked activity in visual cortex[15,22], the pupillary reflex[9,23,47], the ability of behaving animals to discriminate constant light from constant dark[9,11,14,20,22,23,46,47] and a fright response to a looming cue[21], but two-dimensional pattern recognition of the kind needed for patterned vision has not been demonstrated, nor has vision of natural objects. We designed two sets of behavioral tests, the first to determine whether *rd1* mice regain the ability to distinguish between different two-dimensional spatial patterns and the second to determine if they regain the ability to distinguish between different three-dimensional natural objects.

To assess visual recognition of two-dimensional line patterns, we took advantage of the high sensitivity to light of MW-opsin and rhodopsin to display line patterns as visual cues on standard LCD screens. We used an active avoidance task to test the ability of animals to discriminate between patterns of different spatial arrangement and orientation. *Rd1* mice expressing MW-opsin performed, as well as normally sighted *wt* mice with intact rods and cones in distinguishing static parallel lines of vertical versus horizontal orientation. In stark contrast, *rd1* mice expressing rhodopsin were unable to perform this static orientation task. We conjecture that, even when the image is still, saccadic movement of the mouse eye or head blurs the image beyond recognition because of the slow refresh kinetics of rhodopsin. An acuity discrimination task in which mice must distinguish between lines separated by different distances revealed that MW-opsin expressing *rd1* mice perform, as well as *wt* mice, even when the bars are in motion. Accounting for distance from the point of decision and the dimensions of our patterned stimuli, this amounts to a discrimination of ~18° and 0.056 cycles per degree (Supplementary Methods), ~9-fold lower than the resolution of visually intact *wt* mice (~0.3–0.5cpd) reported in earlier behavioral studies[40,55–57].

To assess three-dimensional object vision, we used in an open-field behavior, where mice naturally explore novel objects under normal incidental room light, and where distance to the objects dictates that the major determinant in the behavior is vision. Each of the behavioral measures that were found to depend on vision, i.e., to differ between sighted *wt* animals and blind *rd1* animals (total distance traveled, average velocity of exploration, latency to reach the object, velocity of trajectory to the object, and distance traveled to reach the object) were restored in previously blind *rd1*

animals to *wt* levels. This suggests that MW-opsin in RGCs restores the visual recognition that objects are novel.

Vertebrate vision operates over a very wide range of intensities by mechanisms of adaptation to ambient light[58]. Photoreceptor adaptation is achieved via two main mechanisms: (1) dynamic desensitization of the opsin's G-protein signaling cascade and (2) partial bleaching of retinal pigment during illumination, allowing the fraction of opsin that is sensitive to light to be titrated by ambient light[59,60]. Replenishment is presumably supplied by the retinal pigment epithelium (RPE) or by Müller glia cells located in close proximity to the RGCs and were recently found to contain vitamin A isomerase activity[61]. It seems unlikely that dynamic desensitization would transfer to RGCs with a cone opsin, since nodes of control that mediate light adaptation in the G-protein signaling cascade may be specialized to the photoreceptor cells. We asked whether the light response mediated by MW-opsin in RGCs would adapt to ambient light levels. We found that the light response of the isolated *rd1* retina expressing MW-opsin shifts in sensitivity by >100-fold when changing from dark to light adaptation. The maximal peak response was maintained, meaning that the signal to noise is preserved as the intensity curve shifts from dim indoor to moderate outdoor light levels. Importantly, the light adaptation participated usefully in the behaving animal in a learned visual discrimination task of spatial pattern recognition. The substantial adaptation shift in sensitivity, which, among optogenetic systems for vision restoration, is thus far unique, suggests that MW-opsin could provide patients with a dynamically adjusted vision restoration for indoor and outdoor environments.

The high sensitivity of MW-opsin solves a major challenge of optogenetic gene therapy by eliminating the need for the light intensifying goggles currently used in clinical trials and, therefore, concern about photo-damage to the surviving retina. Compared to the microbial opsins or foreign molecules, restoration in patients using gene delivery of native protein such as MW-opsin reduces the risk of immune reaction or the subsequent need for localized or systemic immune suppression. Remarkably, the vision mediated by MW-opsin displays light adaptation over a range that is suited to vision at both indoor and outdoor light levels. Off-the-shelf adjusting sunglasses could provide a simple solution to expand operation to bright outdoor light. A retinal prosthetic that recapitulates aspects of natural photoreceptor-derived vision in terms of sensitivity, speed and capacity for light adaptation may complement remaining fragments of natural vision in cases of partial, localized, or early stage retinal degeneration. Previous optogenetic therapies have been suited to patients with no light perception. However, the most common forms of blindness, including AMD, maintain peripheral vision but lose the photoreceptors that mediate central, high acuity vision. Delivery of MW-opsin locally to macular and foveal regions may restore useful central vision to these patients.

The unique combination of properties distinguishes MW-opsin for clinical application in patients suffering from a wide range of degenerative retinal diseases that lead to loss of photoreceptor cells. While MW-opsin operates well in white light, it differs from the other cone opsins in wavelength sensitivity, displaying behavioral light avoidance consistent with the medium wave action spectrum. This selectivity holds an exciting potential for future expansion to restoration of color vision once advances in genetic and viral capsid targeting allow different cone opsins to be expressed in specific cell subtypes.

In summary, MW-opsin in RGCs restores to a blind mouse model of retinitis pigmenotosa the ability to recognize visual patterns on an LCD computer display, holding promise for enabling blind patients to read and use video. Moreover, it restores visual function with three-dimensional objects in indoor

light, suggesting that it will support vision during ambulatory activity in patients. The system adapts to light levels between indoor and outdoor illumination, a built-in adaptation that circumvents the need for intensifying goggles. The combination of speed, sensitivity, and adaptation holds promise for vision restoration under natural and changing conditions.

## Methods

**Animals and AAVs.** *Wt* mice (C57BL/6 J) and *rd1* mice (C3H) were purchased from the Jackson Laboratory and housed on a 12-h light/dark cycle with food and water ad libitum. cDNA encoding vertebrate medium wave cone opsin or rhodopsin was linked with yellow florescent protein (YFP) gene on the c-terminus and inserted in an established viral cassette under control of the human synapsin promoter (hsyn-1). Gene and promoter was flanked by inverted terminal repeat domains, stabilized by a polyadenylation signal sequence (polyA) with a woodchuck hepatitis post-transcriptional regulatory element (WPRE) and packaged in the AAV 2/2-4YF capsid. The titer of AAVs was determined via qPCR relative to inverted repeat domains standard and reported to contain $10^{13}$–$10^{14}$ viral genomes. AAVs were produced as previously described[1]. Vector was delivered in a 2 μl volume to the vitreous of the *rd1* mouse eye via microinjection using a blunt 32-gauge Hamilton syringe though an incision made posterior of the ora serrata using a sharp 30-gauge needle. rAAV injections were at p30–p60 and in vivo and in vitro experiments at p90–p160. Mice were anesthetized with IP ketamine (72 mg/kg) and xylazine (64 mg/kg). Eyes were anesthetized with proparacaine (0.5%) and pupils were dilated with phenylephrine (2.5%) and tropicamide (1%).

**Tissue preparation and immunohistochemistry.** Mice >4–6 weeks post-*AAV2/2-hsyn-MW-coneopsin-YFP* treatment were sacrificed, eyes were fixed in 4% paraformaldehyde (Ted Pella) (30 min), retinas were removed and washed thoroughly using PBS and flat mounted on slides using Vectashield (Vector Laboratories) medium impregnated with 4,6-diamidino-2-phenylindole (DAPI) (cell nuclei stain —blue). For retinal sections, whole mounts were embedded in agarose (Sigma) and sectioned transverse using a vibratome (Leica Microsystems) at medium speed, maximum vibration, and 180-μm thickness. Retinal tissues used for immunohistochemistry on retinal cryosections or whole mounts were processed and examined by confocal microscopy (Leica TCS SP5; Leica Microsystems).

**Electrophysiology and light stimulation.** HEK cell recordings we performed using standard electrophysiological techniques previously established[22,62,63]. Briefly, cells were clamped in whole cell mode in a high external potassium solution (50 mM), and held at $V_H = -80$ mV, to provide an inward driving force for potassium. 5–10 s pulses of light were given at low intensity (1 mW cm$^{-2}$) at 535 nm (for MW-opsin) or 500 nm (for rhodopsin). MEA recordings were performed on *wt* (C57BL/6 J) mice, and untreated and treated *rd1* mice at >p90 6–16 weeks following AAV injection experimental retina were excised from the eye under dim red light, mounted on 4 μm cell membranes and placed in an incubator (35 °C) for 30 min and perfused with exogenous chromophore 9-cis retinal within the Ames media. Retinal tissue was placed ganglion cell side down[4] in the recording chamber (pMEA 100/30iR-Tpr; Multi Channel Systems) of a 60-channel MEA system with a constant perfusion of Ames recording media (32 °C). A Multi Channel Systems harp weight (Scientific Instruments—Slice grids) was placed on the retina to prevent movement and vacuum was applied to the retina using a pump (perforated MEA1060 system with CVP; Multi Channel Systems), improving electrode-to-tissue contact and to provide improved signal-to-noise ratios across retinas. Additionally, a dry crystal of exogenous chromophore 9-cis retinal was dissolved in 2 μl 100% ethanol under dark conditions. Dissolved retinal was then added to 100 μl matrigel on ice and added to the top of the harp, just above the retina, in order to provide a proximal reservoir of chromophore (pseudo-RPE). Additionally, 9-cis retinal was dissolved in the recording solution and perfused consistently into the recording chamber. Further detail regarding MEA methods are previously detailed in Gaub et al. (2015)[1]. Illumination in vitro was by a 300-W mercury arc lamp (DG-4; Sutter Instruments) with a 535/50 nm bandpass filter for MW-opsin or a 510/89 nm bandpass filter for rhodopsin. Light intensity was controlled by modifying the light source duty cycle or by using neutral density filters and ranged from 0.02 μW cm$^{-2}$ to 20 mW cm$^{-2}$ (535/50 nm) 0.06 μW cm$^{-2}$ to 57 mW cm$^{-2}$ (510/89 nm). Illumination protocols consisted of 100 ms light flashes at 60 s intervals unless otherwise specified. Contrast experiments on the MEA were performed using an Epson 1040 home theater projector collimated through a shutter and 4× objective so that the entire recording area of the retina was illuminated. Various percentages of gray scale were projected. 100% light = 25 μW cm$^{-2}$. Relative comparisons with natural light intensities were obtained in various environments using direct light measurement with a power meter (Thorlabs) (see Comparing light sensitivity of optogenetic probes, below). Spectral component of white light measured using a CCD Spectrometer with Fourier Transform Optical Spectrum Analyzer software (Thorlabs). See Supplementary Methods for further description. In vivo recordings were performed as described in Supplementary Methods and previously described by Veit et al.[64].

**MEA data acquisition and analysis.** Retinal activity on the MEA was sampled at 25 kHz filtered between 100 and 2000 Hz and recorded using MC_rack software (Multi Channel Systems). Voltage traces were converted to spike trains offline and the spikes recorded at each electrode were sorted into single units, which we defined as "cells," via principal component analysis using Offline Sorter (Plexon-64bit) with each electrode commonly identifying 1–3 cells. Single-unit spike clusters or channels were exported to MATLAB (MathWorks) and were analyzed and graphed with custom software. All firing rates were extracted from traces averaged with 20–50 ms bins over 3–10 light response cycles unless otherwise specified, details of which are denoted in figure legends. Rasters were generated from average firing rates. Spontaneous activity in the degenerate retina was variable within and across samples so thresholds were applied individually so that responses could be identified from spontaneous spiking. Responses across cells and across retina were normalized using the Light Response Index (LRI) adopted from Tochitsky et al.[11] and Gaub et al.[20] (LRI = (peak firing rate in the light−average firing rate in dark)/peak firing rate in the light + average firing rate in dark). Under experiments where conditions were changed within retina (light sensitivity, light and dark adaption, and dependence of response on flash duration etc.) the responses were normalized to the peak of the greatest response from baseline and channels were tracked across all recording parameters. For response recovery of MW-opsin and rhodopsin, peak responses for averaged population of cells within each retina were compared with light responses for 4 sequential flashes and were visualized, extracted and normalized to the first flash. Response sensitivity of MW-opsin and rhodopsin were determined by averaging of 4–5 (at 0.02–60 μW cm$^{-2}$) or 3–2 (at 0.2–6 mW cm$^{-2}$) 100 ms light flashes with 60 s intervals. For our analysis we measured the average peak response of each retina and fitted the fast decay component with a single exponential. All curve fitting and kinetic analysis was performed in Clampfit 10.6 (Molecular Devices). Cells were defined as "responders" if the LRI satisfied the condition LRI >0.1. The width of response at half maximum of peak from baseline was determined manually using Clampfit 10.6 measurements. Intensity-response relations were fit with a single Boltzmann and normalized to the fit between 0 and 1.

**Tissue preparation and immunohistochemistry.** Mice were sacrificed >4–6 weeks after intravitreal injection of *AAV2/2-hsyn-MW-coneopsin-YFP* treatment. Eyes were isolated and fixed in 4% paraformaldehyde (Ted Pella) (30 min) and either flat mounted or embedded in agarose (Sigma) and sectioned in transverse using a vibratome (Leica Microsystems), as done previously[19,20,22]. Retinal tissues were examined by confocal microscopy (Leica TCS SP5; Leica Microsystems). See Supplementary Methods for further description.

**Statistics.** To assess statistical significance of MEA recordings, nonparametric two-tailed Mann–Whitney $U$ tests where applied. For learned dark avoidance behavior and the learned pattern discrimination behaviors significance was determined in two ways. (1) Significance for behavioral performance was calculated using two-tailed unpaired student's t-tests with Bonferroni correction when applicable. Significance was also determined by computing the proportion of successful performances (2). A success was defined as greater than the sum of the control group average and standard deviation. Success ratios were then calculated for each condition and graphed in the Supplementary Figures as proportion of avoidance. To determine significance in differences between conditions a pairwise contingency table was then constructed, and a Two-Sided Pearson's Chi-Square Test was initially conducted. To correct for any conditions with a small n, a One-Sided Fisher's Exact Test was also conducted (Supplementary Table 1).

**Behavioral analyses.** The 2-chamber light-dark passive avoidance test was performed as described previously[19,20,22,23]. White light (wavelength range) at ~100 μW cm$^{-2}$ or either blue light (460/45 nm) or green light (535/50 nm) at 0.5–25 μW cm$^{-2}$ was mounted above the chamber with homogeneously distributed light. Animal movements were tracked using IR sensors on the shuttle box. Time spent in the light and dark chambers was measured and analyzed using Graphic State and Graphic State RT (Coulbourn Instruments).

Fear conditioning experiments were performed as described previously[22] using Coulbourn single shock chamber with an LED screen that presented the visual cue mounted to the ceiling of the chamber. Animals were subjected to paired or unpaired light cued fear conditioning consisting of three shock trials at 0.7 mA over a span of 15 min. Freezing behavior in anticipation of the shock was recorded by Coulbourn's FreezeFrame software and normalized to movement behavior gathered before the stimulation. Performance was compared between paired and unpaired cohorts in order to determine if a fear response was conditioned to the stimulus transition.

Modified active avoidance was assayed as previously described[19], using the Coulbourn shuttle box (H10-11M-SC), however, now iPad tablet screens were mounted onto the shuttle cage wall, each displaying one of two images that differed in orientation or distance between two lines but were otherwise of equal shape, size, and light intensity. The aversive image side was paired with a foot shock of 0.7 mA. Upon recall the light patterns were reversed to avoid a bias for location and time spent on each side was recorded. Adaptation was tested by dimming or brightening the display to different intensities. Visual discrimination optical angle calculations

were performed using the parameters of the behavioral shuttle cage ($15.24 \times 36$ cm), the distance from the decision point (divider), the central position of the LCD panel (18.85 cm), and the parameters of the stimulus pattern (1–6 cm distance between the parallel lines) using the optical (physical) angle equation. Visual angle = V = $2\tan^{-1}((D/2)/(L)) = 18$ degrees = 0.33–0.49 radians. Cycles per degree = $1/V \sim 0.056$ cpd. This is ~9-fold lower than performance in visually intact *wt* mice (~0.3–0.5cpd) reported in other studies[40,55–57].

For exploratory behavior analysis, two objects were placed in a $50 \times 50$ cm open field box. Animals were positioned in the empty box and allowed to explore freely over the course of 10 min. The following day, two novel objects were placed in the box and animals were again positioned along the wall of the box and allowed to explore freely for 10 min while the arena was filmed continuously. Using Noldus Technology Ethosvision XT v13.5, videos were analyzed for distance traveled (cm), the velocity of travel (cm/s), the latency (s), velocity (cm/s), and distance traveled (cm) to arrive at and explore each object.

For detailed description of behavioral analysis methods see Supplementary Materials.

**Reporting summary**. Further information on experimental design is available in the Nature Research Reporting Summary linked to this article.

## Data availability
The data that support the findings of this study are available from the corresponding author upon reasonable request.

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

## Acknowledgements

We thank Hillel Adesnik, Leah Byrne, Prashant Donthamsetti, Cameron Baker, Tim Day, Robert Duvoisin, and Richard Kramer for helpful discussion and Leah Byrne, Aaron Friedman, Yang Joon Kim, Adam Hoagland, Zhu Fu, Cherise Stanley, Aleksandra Polosukhina, and Ivan Tochitsky for technical assistance. The work was supported by the National Institutes of Health (NEI) Nanomedicine Center for the Optical Control of Biological Function 2PN2EY018241 (E.Y.I. and J.G.F.) and EY024958 and EY022975 (J.G.F.), and the Foundation Fighting Blindness, USA, Hope for Vision Foundation, The Lowy Medical Research Institute (J.G.F.) and by an Achievement Rewards for College Scientists (ARCS) Foundation scholarship (M.H.B.).

## Author contributions

M.H.B., A.H., J.L., J.V., A.S., K.A., B.S. and B.G. designed and performed experiments and analyzed data, with input from J.G.F. and E.Y.I. H.E.K. cell recordings were performed by J.L., Multielectrode array recordings by M.H.B. and A.S., Cortical recordings by J.V., behavioral experiments by A.H., and viral synthesis and cloning by M.V. The paper was written by M.H.B. and E.Y.I. with input from all of the authors.

## Additional information

**Competing interests:** J.G.F and E.Y.I. are founders of Photoswitch Therapeutics, a startup whose goal is to restore vision in blinding disease using photo-pharmacology, chemical optogenetics, and optogenetic methods of the kind described here. The remaining authors declare no competing interests.

