## [Peer Review File · Nature Communications]

Reviewers' comments:

Reviewer #1 (Remarks to the Author):

In their study, Berry et al. test to what extent adeno-associated virus (AAV-) mediated expression of medium wavelength sensitive cone (MW-) opsin in retinal ganglion cells restores visual function. A previous study discovered that cone opsins could drive light responses in non-photoreceptor neurons and that these responses have faster kinetics than those mediated by rhodopsin (Masseck et al. Neuron 2014). That MW-opsin can drive light responses of retinal ganglion cells and that these responses support vision at higher spatiotemporal frequencies than rhodopsin expression in retinal ganglion cells is, therefore, not particularly surprising. Given the therapeutic potential of this approach, which improves vision restoration compared to previous methods in meaningful ways, this study, nonetheless, still is potentially appropriate for publication in Nature Communications. Toward this end, the authors should improve their characterization of light response properties of retinal ganglion cells expressing MW-opsin to allow for a more comprehensive comparison to other approaches. Also, they need to address concerns/comments about some of their behavioral assays.

Specific comments

- 1.) The authors should characterize light responses of retinal ganglion cells expressing MW-opsin more thoroughly to reveal how visual information differs between wild-type, MW-opsin, and rhodopsin-expressing cells. They should measure (a) spatiotemporal frequency tuning (e.g., by presenting drifting grating stimuli) and (b) contrast sensitivity (e.g., via drifting gratings or full-field steps).
- 2.) In retinal degeneration, spontaneous hyperactivity in the inner retina contributes to the disruption of vision. The authors should show whether spontaneous hyperactivity persists in MW-opsin expressing retinal ganglion cells and how light affects spontaneous oscillations of these, wild-type and rhodopsin-expressing cells.
- 3.) The authors use the placement of the central divider in the behavior box illustrated in Figure 3f to estimate acuity. I do not think this is justified. Mice could get much closer to the LCD at the end of the box to identify the stimulus pattern and choose whether to stay or leave the respective side of the box. The authors could place transparent barriers on each side to establish a minimal viewing distance that would allow them to estimate a lower bound of acuity.
- 4.) In Figure 5 the authors analyze exploratory behavior. Rd1 mice move more slowly and, overall, travel less. To test whether mice that express MW-opsin in their ganglion cells chose a more direct path to visible objects than rd1 mice and to evaluate how they compare to wild-type, I think it might be useful to show the distance traveled per object encounters.

Reviewer #2 (Remarks to the Author):

This ms by Berry and colleagues tackles a long-standing problem in the field of visual restoration: how to confer light sensitivity to retinas devoid of photoreceptors in a manner that allows visual perception across the normal, wide range of light levels encountered daily? Here, the authors report that ectopic expression of MW-opsin in the rd1 retina achieves sensitivity equal to that of rhodopsin but with improved adaptation and kinetic properties that restores visually-guided behavior at relatively light levels.

Although there has been a spate of visual restoration papers similar to this one, each improving upon the last, this ms is a particularly notable and important advance because they demonstrate a) functionality over a 1000-fold range of light levels and b) improved temporal resolution that

appears to permit form vision, at least in the mouse. There is little doubt this paper will inspire many labs to try this approach in their own models.

Areas for improvement:

1. The introduction does not provide adequate background, with 15 of the 38 lines describing the work to be presented. It would be more useful to get specific about what has been achieved by the other methods to date, and what range values of sensitivities and kinetics are most needed to advance the field? e.g. scholarly discussion of Supplemental Fig 2. It would also be appropriate to describe the known structural and functional differences between rhodopsin and cone opsins (e.g. Imamoto and Shichida, 2014 *Biochim Biophys Acta* 1837, 664-73) and why M-opsin is such a logical choice.

2. The authors note that in response to the brightest flash, dark-adapted responses recovered much more slowly than under light-adapted conditions. Such a profound difference was not observed for dimmer flashes (Fig. 4a). The authors state this is similar to the behavior of photoreceptors and believed to play a role in desensitization. Since they have no evidence that this anomaly has anything to do with photoreceptors (and none exist in the rd1 retina), it would be better to remove this sentence and merely explain that because of its unusual properties the aberrantly slow response was excluded.

3. Fig. 4c plots the light response index (LRI) - please define LRI in the main text, not only the Supplement. The text (L223) refers to Fig 4C as evidence that the maximal light response was similar in dark- and light-adapted retina, but I can't see that from the data presented in this panel.

4. It was difficult to understand the light avoidance experiment of Fig. 4e, so the authors might want to consider editing the middle paragraph on p8 for clarity. The dark-adapted mice spent roughly 550 s in the dark vs 1 $\mu\text{W cm}^{-2}$. The light-adapted mice spent less time in the dark vs 1 $\mu\text{W cm}^{-2}$ (Fig. 4e), consistent with the mice not being able to tell the difference between dark and dim light. However, there was a substantial increase in the time in the dark vs 100-fold brighter light (100 $\mu\text{W cm}^{-2}$). How do the authors explain the fact that neither 1 nor 100 $\mu\text{W cm}^{-2}$ was sufficient to generate a significant change in firing rate (Fig. 4b)? Convergence in the LGN?

5. The most compelling demonstration of the consequences on MW-opsin expression is the restoration of visually guided exploratory behavior. It is truly remarkable that the expression restored all of the measured parameters to WT levels. Were there any parameters that the expression did not affect? Is it possible that the treatment itself (anesthesia, transient immune response) alters the degree of animal's vigilance, physical activity or attention? How do rhodopsin-expressing rd1 mice do?

6. There are several instances where the authors inaccurately report photoreceptor properties or incorrectly attribute their observations to them:

a. The authors point to the faster decay of light responses in cones compared to rods to explain the rhodopsin vs cone opsin differences in their experiments (e.g. L294-95). This is incorrect and the references cited here (refs 49, 50, 51) don't claim this either. Rods are slow because of slow G-protein deactivation (e.g. Krispel et al., 2006). However, there are intrinsic differences in the active lifetime, gain, regeneration rates, and thermal stability of rhodopsin vs cone opsins, and these fundamental differences in the opsins are what is relevant in this context. See for example Imamoto and Shichida, 2014 *Biochim Biophys Acta* 1837, 664-73.

b. Discussion of photoreceptor adaptation mechanisms (L343-345). The loss of quantum catch from depletion of available pigment molecules - so-called "bleaching adaptation" - is a feature only of cone adaptation. The references cited here (39,40) where Pepperberg is describing bleaching desensitization in rods is a completely different mechanism involving the accumulation of photoproducts with partial ability to activate the downstream cascade. Bleaching adaptation (pigment depletion) goes all the way back to Alpern (1970) but is somewhat better summarized

for M-cones in Mollon and Polden (1977) *Nature* 265, 243-246 or Shevell (1977) *Vision Research* 17, 427-434.

c. Light-adapted cones and the maximal response amplitudes (L223): It is not correct that the maximal response amplitude of cones does not vary with background intensity. They do not saturate (there is always some residual light-suppressible current in most cones) but the amount of suppressible current (maximal amplitudes) decreases substantially. Neither of the citations - the review by Hurley (ref 42) or paper by Normann and Werblin (ref 43) state this. The comparison is unnecessary anyway - I suggest to simply delete.

7. L328-331. Please justify the conversion of visual angle into Snellen values in this context. The same visual angle covers a much larger distance on the human retina. Was the difference in mouse vs human eyes accounted for after using the conversion chart of ref 52? Please provide more detail (show the entire calculation with equations in the Supplement) or else please delete this. Others will mimic and false claims will pervade.

Minor:

Typo: L178 refers to slow response kinetics of rhodopsin-driven MEA responses in Fig. 3a, e rather than Fig. 2a,e.

It is curious that the Tpeak is considerably longer than the measured Tau OFF (Fig. 2D). This suggests that the onset and offset of the response are driven by distinct mechanisms. It might be worth pointing out in the discussion that further improvements to the temporal resolution should include speeding up onset.

Reviewer #3 (Remarks to the Author):

Berry et al. report that the medium wavelength sensitive (MWS) cone opsin, when expressed ectopically in retinal ganglion cells (RGCs), can render these cells photosensitive. The artificial RGC response has a unique combination of features that is attractive for vision restoration. It is sensitive enough to function under natural lighting conditions. It is adaptive, such that it has a broad dynamic range. And it is rapid, allowing for temporal (and thus spatial) resolution. Being naturally expressed in the eye, overexpression is unlikely to be harmful. Indeed, when the authors transduce rodless/coneless mice with MWS opsin, these animals gain the ability to perform various visual tasks, even those that require pattern discrimination.

This manuscript comes from a group that has led the way on molecular strategies for vision restoration. On the whole, the experiments are well-designed, the data are of high quality, and the manuscript is clearly written. I have two principal critiques. One is that the information obtained is not readily generalizable. Another is that signaling by MWS opsin may be too fleeting to be useful. If these concerns were addressed, this manuscript would be a fine candidate for *Nature Communications*.

Major Comments

1. Temporal resolution of the MWS response

The authors characterize the waveforms of MWS responses that are evoked by rather arbitrary stimuli (i.e., pulses that are generally too long to produce impulse responses per se). These data are enough for a comparison between MWS and rhodopsin responses, but do not allow generalization to other stimulus conditions.

A more comprehensive understanding of MWS response kinetics would be valuable. It could be obtained by measuring the flicker fusion frequency, or by performing a cross-correlation of the response with a white noise stimulus. Ideally, this could be done under photopic conditions when

temporal resolution is highest (assuming that response kinetics accelerate with light adaptation as in the normal photoreceptors). On a related note, how much heterogeneity is there across RGCs?

2. Adaptation of MWS responses

The authors show that conditioning with moderate illumination (10 min of 0.1 mW/cm²) causes the sensitivity of MWS responses to shift such that threshold is near 10 mW/cm². Thus, it would appear that the retinas have entirely lost their responses to moderate light intensities by the end of the conditioning period. This is the case even with a continuous supply of 9-cis retinal. By extension, an MWS animal would have, at most, 10 min of useful vision when going from darkness to a moderately lit room. This is a severe limitation on LWS-mediated vision restoration.

It would be important to examine the responses of MWS-expressing RGCs under continuous illumination at different intensities. To what extent do they decline, and with what time course? Is there a particular range of light intensities over which MWS responses are sustained—that is, where the rate of bleaching/adaptation does not outpace that of regeneration? What is the time course of recovery from adaptation after illuminations of different intensities cease?

To provide a more generalizable understanding of adaptation, the authors should consider measuring sensitivity as a function of continuous background light of different intensities (not the first and in cones, but Matthews H...Lamb 1990 J Physiol provides a nice illustration of such an experiment). If the kinetics of the flash response changes with background light, that is a sign that adaptation results not just from bleaching but also involves other mechanisms of negative feedback, which would be interesting.

Additional Comments

1. The authors state that the sensitivities of cells expressing rhodopsin and MWS cone opsin are similar, but this is not entirely true. Their sensitivity to brief stimulation is similar, but not to steady stimulation—with prolonged illumination, the long duration of the rhodopsin response will allow for temporal summation and thus a higher effective sensitivity than the MWS response. It would be helpful to be clearer on this point.

2. The MWS response clearly has more than one decay phase; some quantification of the slow phase, and a discussion of how it impacts temporal resolution, is warranted.

3. Fig. 2f and g. Does the same relation hold when delivering pulses from shortest to longest duration; that is, in reverse order? Alternatively, is there some assurance that the responses are recovering completely between pulses? From the data shown, each response appears to return to baseline, but it is unclear if sensitivity does.

4. How consistent is the extent of transduction across animals and retinas?

5. What fraction of RGCs detected on the array are photosensitive following transduction?

6. Why are the intensity-response curves different between Fig. 2b and 4b (the dark-adapted curve)? The latter saturates at a much lower intensity and might also be right-shifted.

7. Throughout the paper, it appears that stimuli of different wavelengths are matched by energy and not photon number, even though it is the latter that is important for photoreception. There should be some discussion of whether this affects the comparisons being made. I would encourage the authors to use photon number in future studies.

8. How were comparisons among various opsins made (e.g., the ChR2 minimum in Fig. 4c, Fig. S2)? Is the threshold for each measured at its maximum wavelength of absorption? Were the

values reported in the various studies for bandpass filters of similar width, using stimuli that were of comparable duration (because longer stimuli afford more opportunity for temporal integration)? When the experiments were done in vivo, were the pupils dilated, and were expression levels comparable across studies? These comparisons are important to make, but they are complex and should be discussed in a transparent and nuanced fashion.

9. On a related note, it would be helpful to add melanopsin to Fig. S2 if possible.

10. In the discussion of how MWS opsin in RGCs would be regenerated in vivo (lines ~350) the authors might consider citing the literature on the visual cycle in Muller glia.

11. Line 299: "MW opsin is fast enough to respond to a flash of light as brief as 25 ms." This does not quite make sense. The key variable here is whether there are enough photons delivered that enough are absorbed to produce a detectable response. The duration of delivery is practically immaterial--visual pigments activate on the femtosecond time scale.

12. Line 328, it would be helpful to express acuity in terms of cycles per degree.

13. The manuscript should be proofread for typos (e.g., "Coulbourn") and grammatical errors.

We thank the reviewers for their helpful comments. We have addressed all of the questions that were raised, as described below. To do so, we added new experiments and new analysis, and expanded our introduction and discussion of the results (changes to text highlighted yellow in the main text and supplementary text). These new results are documented in a series of new supplemental figures, as listed below. Two researchers (Julia Veit and Benjamin Sivyer) contributed critically to the new experiments and have been added as authors..

Figure 5 adds a comparison with rhodopsin expressed in RGCs of the *rd1* mouse to our characterization of restoration of visually-guided exploratory behavior by MW-opsin expressed in RGCs of the *rd1* mouse as well as new quantitative measures of the behavior (velocity and distance to first and second objects).

Figure S2 documents rhodopsin expression in RGCs of the *rd1* mouse retina

Figure S3 quantifies the number of active RGCs picked up in our MEA recordings from the *rd1* mouse retina expressing MW-opsin in RGCs and compares this to responses measured in the wildtype retina.

Figure S4 quantifies the properties of the slow component of the light response of the *rd1* retina expressing MW-opsin in RGCs.

Figure S6d and e compare the kinetics of the light response in RGCs of wildtype retina and *rd1* retina expressing MW-opsin in RGCs.

Figure S7 shows contrast detection of isolated *rd1* retina expressing MW-opsin in RGCs.

Figure S8 shows *in vivo* light responses in the primary visual cortex of awake, free walking *rd1* mice expressing MW-opsin with multi-electrode recordings of visual evoked potentials.

Figure S9 shows contrast detection in V1 layer 4 cells measured *in vivo*.

Figure S10 shows temporal properties of visually evoked responses in the V1 of awake, free walking *rd1* mice expressing MW-opsin in RGCs.

Figure S12 shows the location preference within the discrimination task behavioral apparatus, and reveals the decision point that can be used to calculate visual acuity.

Figure S13g shows light adaptation in visual behavior and reveals that MW-opsin expressing *rd1* mice can perform the task even after 8 hours of light adaptation, indicating that the light adapted state is not simply an absorbing bleached state that will blind animals soon after they are exposed to ambient light, but a dynamically regulated state that can support vision for extended times.

Reviewer #1

In their study, Berry et al. test to what extent adeno-associated virus (AAV-) mediated expression of medium wavelength sensitive cone (MW-) opsin in retinal ganglion cells restores visual function. A previous study discovered that cone opsins could drive light responses in non-photoreceptor neurons and that these responses have faster kinetics than those mediated by rhodopsin (Masseck et al. Neuron 2014). That MW-opsin can drive light responses of retinal ganglion cells and that these responses support vision at higher spatiotemporal frequencies than rhodopsin expression in retinal ganglion cells is, therefore, not particularly surprising. Given the therapeutic potential of this approach, which improves vision restoration compared to previous methods in meaningful ways, this study, nonetheless, still is potentially appropriate for

publication in Nature Communications. Toward this end, the authors should improve their characterization of light response properties of retinal ganglion cells expressing MW-opsin to allow for a more comprehensive comparison to other approaches. Also, they need to address concerns/comments about some of their behavioral assays.

Specific comments

1.) The authors should characterize light responses of retinal ganglion cells expressing MW-opsin more thoroughly to reveal how visual information differs between wild-type, MW-opsin, and rhodopsin-expressing cells. They should measure (a) spatiotemporal frequency tuning (e.g., by presenting drifting grating stimuli) and (b) contrast sensitivity (e.g., via drifting gratings or full-field steps).

In order to provide more through characterization, contrast sensitivity using full-field gray scale steps were measured on the multi-electrode array (MEA) in the excised retina of *rd1* mice expressing MW-opsin in RGCs. The responses were compared to recordings under the same conditions from wildtype retina. The light stimulation was delivered by a standard projector (Epson, model 1040), zeroed to a black background that stepped to various levels of gray scale. We find that MW-opsin reliably changes activity in response to changes in light level of 25% (Fig. S7a,b). This performance approaches but does not equal that of wildtype retina, which was sensitive to smaller changes in light level (Fig. S7c).

Having seen the contrast sensitivity of MW-opsin in the isolated retina, we wanted to know how MW-opsin operates in visual performance. To assess this, we added a new class of experiment to our study: *in vivo* recordings of visually-evoked potentials (VEPs) in the primary visual cortex of the awake, free running *rd1* mouse expressing MW-opsin in RGCs. Recordings were made with a 16-channel linear electrode array, with electrodes at 25 micron spacing, allowing us to record single units across the layers of cortex in response to changes in the brightness of a standard LCD computer monitor that was placed in front of the mouse in such a way that it occupied a large portion of the visual field (Fig. S8). We first tested contrast sensitivity *in vivo* by measuring activity as light intensity was changed randomly. As seen in the isolated retina, substantial changes in firing of layer 4 neurons were seen in response to increases in brightness of 15% or more (Fig. S9). We next examined temporal resolution *in vivo*. Full-field flickering stimuli were presented and we observed that the activity of layer 4 neurons reliably followed frequencies of 0.5, 1, 2, and 4 Hz (Fig. S10).

These *in vivo* experiments are demanding and took some time to set up, so that in the 3 months since we received the reviews we were unable to perform the drifting grating experiment. This experiment, and others that would create a full characterization of response properties, would require 1-2 years of further study. But we can already conclude that the contrast sensitivity and temporal properties of the light responses that we see in the MEA recordings from the isolated retina are conveyed to cortex and therefore available to the visual system.

2.) In retinal degeneration, spontaneous hyperactivity in the inner retina contributes to the disruption of vision. The authors should show whether spontaneous hyperactivity persists in

MW-opsin expressing retinal ganglion cells and how light affects spontaneous oscillations of these, wild-type and rhodopsin-expressing cells.

We did not see a systematic difference in spontaneous activity in the dark between *rd1* retinas that were untreated and ones that expressed MW-opsin in RGCs.

However, we did observe that *rd1* retinas expressing MW-opsin in RGCs had a transient suppression of spontaneous activity for several seconds following a light pulse, as shown below. This property of the MW-opsin light response in RGCs may enhance the signal to noise and may be part of the mechanism of adaptation to changes in ambient light level. Because of the complexity of this process we were unable to characterize it quantitatively across conditions of adaptation and light/dark rearing. A future study will be required to do this subject justice.

3.) *The authors use the placement of the central divider in the behavior box illustrated in Figure 3f to estimate acuity. I do not think this is justified. Mice could get much closer to the LCD at the end of the box to identify the stimulus pattern and choose whether to stay or leave the respective side of the box. The authors could place transparent barriers on each side to establish a minimal viewing distance that would allow them to estimate a lower bound of acuity.*

While it is possible for the mice to approach the LCD screen, the mice spend most of their time in the half of each chamber that is close to the central divider between the two chambers (not near the screen). This is shown in new Figure S12. The reason for this may be that during the two days of training (conditioning phase) the mice receive a foot shock every few seconds whenever they venture into the chamber displaying the adverse stimulus, an experience that may make them favor proximity to the door in the central divider that is the passage between the aversive and non-aversive chambers.

4.) *In Figure 5 the authors analyze exploratory behavior. Rd1 mice move more slowly and, overall, travel less. To test whether mice that express MW-opsin in their ganglion cells chose a more direct path to visible objects than rd1 mice and to evaluate how they compare to wild-type, I think it might be useful to show the distance traveled per object encounters.*

We have expanded our analysis of the exploratory behavior to include: total distance and average velocity traveled, distance and average velocity to first and second objects, and latency to each object. Our comparisons were expanded to include *rd1*-sham injected, *rd1* untreated, *rd1* expressing rhodopsin, *rd1* expressing MW-opsin, and *wt*. These additions are incorporated into a revised Figure 5.

Reviewer #2

This ms by Berry and colleagues tackles a long-standing problem in the field of visual restoration: how to confer light sensitivity to retinas devoid of photoreceptors in a manner that allows visual perception across the normal, wide range of light levels encountered daily? Here, the authors report that ectopic expression of MW-opsin in the rd1 retina achieves sensitivity equal to that of rhodopsin but with improved adaptation and kinetic properties that restores visually-guided behavior at relatively light levels.

Although there has been a spate of visual restoration papers similar to this one, each improving upon the last, this ms is a particularly notable and important advance because they demonstrate a) functionality over a 1000-fold range of light levels and b) improved temporal resolution that appears to permit form vision, at least in the mouse. There is little doubt this paper will inspire many labs to try this approach in their own models.

Areas for improvement:

1. The introduction does not provide adequate background, with 15 of the 38 lines describing the work to be presented. It would be more useful to get specific about what has been achieved by the other methods to date, and what range values of sensitivities and kinetics are most needed to advance the field? e.g. scholarly discussion of Supplemental Fig 2. It would also be appropriate to describe the known structural and functional differences between rhodopsin and cone opsins (e.g. Imamoto and Shichida, 2014 Biochim Biophys Acta 1837, 664-73) and why M-opsin is such a logical choice.

The Introduction and Discussion have been expanded to include a brief summary of what has been achieved using optogenetics, the shortcomings of the approaches to date, and the aspects of human vision that one strives to reconstitute. We now address the differences between cone and rod phototransduction in the Discussion. In addition, a new section entitled “Comparing light sensitivity of optogenetic probes” has been added to Supplemental Materials and Methods, which explains the sensitivity measurement equivalents described in the text and figures, (notably original Fig. S2, currently Fig. S5). The legend to Figure S5 now contains the experimental parameters (animal model, wavelength, AAV etc.) used for each of the sensitivity thresholds reported for the optogenetic approaches in Figure S5.

2. The authors note that in response to the brightest flash, dark-adapted responses recovered much more slowly than under light-adapted conditions. Such a profound difference was not observed for dimmer flashes (Fig. 4a). The authors state this is similar to the behavior of photoreceptors and believed to play a role in desensitization. Since they have no evidence that this anomaly has anything to do with photoreceptors (and none exist in the rd1 retina), it would

be better to remove this sentence and merely explain that because of its unusual properties the aberrantly slow response was excluded.

The sentence was removed.

3. Fig. 4c plots the light response index (LRI) - please define LRI in the main text, not only the Supplement.

Now defined in main text results when describing Figure 1j.

The text (L223) refers to Fig 4C as evidence that the maximal light response was similar in dark- and light-adapted retina, but I can't see that from the data presented in this panel.

The light response (LRI) for the light-adapted retina was 38.2 mW/cm^2 , similar to the maximal LRI of the dark-adapted retina that was evoked at 0.382 mW/cm^2 . To avoid confusion we have removed this point from our description of the result.

4. It was difficult to understand the light avoidance experiment of Fig. 4e, so the authors might want to consider editing the middle paragraph on p8 for clarity. The dark-adapted mice spent roughly 550 s in the dark vs $1 \mu\text{W cm}^{-2}$. The light-adapted mice spent less time in the dark vs $1 \mu\text{W cm}^{-2}$ (Fig. 4e), consistent with the mice not being able to tell the difference between dark and dim light. However, there was a substantial increase in the time in the dark vs 100-fold brighter light ($100 \mu\text{W cm}^{-2}$). How do the authors explain the fact that neither 1 nor $100 \mu\text{W cm}^{-2}$ was sufficient to generate a significant change in firing rate (Fig. 4b)? Convergence in the LGN?

The light intensity that we deliver to the isolated retina is accurately determined because it is directly projected onto the retina, whereas in the live animal the light delivered to the eye is focused by the lens, generating greater intensity per unit area on the retina. In addition, as pointed out by the reviewer, there are several points of convergence in the visual system, including the LGN, which could amplify responses in higher visual areas.

*5. The most compelling demonstration of the consequences on MW-opsin expression is the restoration of visually guided exploratory behavior. It is truly remarkable that the expression restored all of the measured parameters to WT levels. Were there any parameters that the expression did not affect? Is it possible that the treatment itself (anesthesia, transient immune response) alters the degree of animal's vigilance, physical activity or attention? How do rhodopsin-expressing *rd1* mice do?*

We have expanded our analysis of the exploratory behavior to include: total distance and average velocity traveled, distance and average velocity to first and second objects, and latency to each object. In addition, our comparisons were expanded to include *rd1*-sham injected, *rd1* untreated, *rd1* expressing rhodopsin, *rd1* expressing MW-opsin, and *wt*. These additions are incorporated into an expanded Figure 5. As can be seen, not all of the parameters differed statistically, but the salient differences showed rescue of efficient navigation to the novel objects in *rd1* mice only

when they were injected with AAV encoding the MW-opsin and not rhodopsin or the sham injection.

6. *There are several instances where the authors inaccurately report photoreceptor properties or incorrectly attribute their observations to them:*

a. *The authors point to the faster decay of light responses in cones compared to rods to explain the rhodopsin vs cone opsin differences in their experiments (e.g. L294-95). This is incorrect and the references cited here (refs 49, 50, 51) don't claim this either. Rods are slow because of slow G-protein deactivation (e.g. Krispel et al., 2006). However, there are intrinsic differences in the active lifetime, gain, regeneration rates, and thermal stability of rhodopsin vs cone opsins, and these fundamental differences in the opsins are what is relevant in this context. See for example Imamoto and Shichida, 2014 Biochim Biophys Acta 1837, 664-73.*

We thank the reviewer for catching this. The comparison of rod and cone opsins has been corrected and is now only in the Discussion.

b. *Discussion of photoreceptor adaptation mechanisms (L343-345). The loss of quantum catch from depletion of available pigment molecules - so-called "bleaching adaptation" - is a feature only of cone adaptation. The references cited here (39,40) where Pepperberg is describing bleaching desensitization in rods is a completely different mechanism involving the accumulation of photoproducts with partial ability to activate the downstream cascade. Bleaching adaptation (pigment depletion) goes all the way back to Alpern (1970) but is somewhat better summarized for M-cones in Mollon and Polden (1977) Nature 265, 243-246 or Shevell (1977) Vision Research 17, 427-434.*

We thank the reviewer for the correction and these key citations. The text has been corrected and citations have been updated.

c. *Light-adapted cones and the maximal response amplitudes (L223): It is not correct that the maximal response amplitude of cones does not vary with background intensity. They do not saturate (there is always some residual light-suppressible current in most cones) but the amount of suppressible current (maximal amplitudes) decreases substantially. Neither of the citations - the review by Hurley (ref 42) or paper by Normann and Werblin (ref 43) state this. The comparison is unnecessary anyway - I suggest to simply delete.*

We have followed the reviewer suggestion and deleted this.

7. *L328-331. Please justify the conversion of visual angle into Snellen values in this context. The same visual angle covers a much larger distance on the human retina. Was the difference in mouse vs human eyes accounted for after using the conversion chart of ref 52? Please provide more detail (show the entire calculation with equations in the Supplement) or else please delete this. Others will mimic and false claims will pervade.*

Distance from stimulus to decision point (L) = 18 cm

Distance between the lines (D) = 6 cm

Visual angle (V) = $2 \tan^{-1}((D/2)/(L)) = 18 \text{ degrees} = 0.33 \text{ radians}$

Cycles per degree = $1/V \sim 0.056$ cpd

This is ~9-fold lower than performance in visually intact *wt* mice (~0.3-0.5cpd) that was reported in earlier studies ^{1, 2, 3, 4}.

The difference in size between mouse and human eye means that the distance from the nodal point of the eye to retina is far smaller in the mouse. This results in a substantially smaller image size projected to the retina, suggesting that the visual equivalent in human vision may be superior to the angle equivalent predicted by Snellen. However, this is reliant on numerous other factors such as the curvature and contractibility of the lens. These parameters remain unknown in previously blind *rd1* mice with optogenetic restoration. For this reason comparison between the visual angle in mouse and human is indeed challenging. To avoid this, the text has been edited to simply state the cycles per degree and its comparison with the *wt* mouse.

1. Leinonen H, Tanila H. Vision in laboratory rodents-Tools to measure it and implications for behavioral research. *Behav Brain Res*, (2017).
2. Prusky GT, West PW, Douglas RM. Behavioral assessment of visual acuity in mice and rats. *Vision Res* **40**, 2201-2209 (2000).
3. Shi C, *et al.* Optimization of Optomotor Response-based Visual Function Assessment in Mice. *Sci Rep* **8**, 9708 (2018).
4. Wong AA, Brown RE. Visual detection, pattern discrimination and visual acuity in 14 strains of mice. *Genes, Brain and Behavior* **5**, 389-403 (2006).

Minor:

Typo: L178 refers to slow response kinetics of rhodopsin-driven MEA responses in Fig. 3a, e rather than Fig. 2a,e.

Corrected

It is curious that the Tpeak is considerably longer than the measured Tau OFF (Fig. 2D). This suggests that the onset and offset of the response are driven by distinct mechanisms. It might be worth pointing out in the discussion that further improvements to the temporal resolution should include speeding up onset.

A point has been added to the Discussion (Lines 372-3) regarding this.

Reviewer #3

Berry et al. report that the medium wavelength sensitive (MWS) cone opsin, when expressed ectopically in retinal ganglion cells (RGCs), can render these cells photosensitive. The artificial RGC response has a unique combination of features that is attractive for vision restoration. It is sensitive enough to function under natural lighting conditions. It is adaptive, such that it has a broad dynamic range. And it is rapid, allowing for temporal (and thus spatial) resolution. Being naturally expressed in the eye, overexpression is unlikely to be harmful. Indeed, when the authors transduce rodless/coneless mice with MWS opsin, these animals gain the ability to

perform various visual tasks, even those that require pattern discrimination.

This manuscript comes from a group that has led the way on molecular strategies for vision restoration. On the whole, the experiments are well-designed, the data are of high quality, and the manuscript is clearly written. I have two principal critiques. One is that the information obtained is not readily generalizable. Another is that signaling by MWS opsin may be too fleeting to be useful. If these concerns were addressed, this manuscript would be a fine candidate for Nature Communications.

Major Comments

1. Temporal resolution of the MWS response

The authors characterize the waveforms of MWS responses that are evoked by rather arbitrary stimuli (i.e., pulses that are generally too long to produce impulse responses per se). These data are enough for a comparison between MWS and rhodopsin responses, but do not allow generalization to other stimulus conditions.

While some of our light pulses are long, as pointed out by the reviewer, we also use pulses as short as 25 ms in our recordings from isolated retina (Fig. 2f, g) and 50 ms in our new *in vivo* cortical recordings (Fig. S10).

A more comprehensive understanding of MWS response kinetics would be valuable. It could be obtained by measuring the flicker fusion frequency, or by performing a cross-correlation of the response with a white noise stimulus. Ideally, this could be done under photopic conditions when temporal resolution is highest (assuming that response kinetics accelerate with light adaptation as in the normal photoreceptors). On a related note, how much heterogeneity is there across RGCs?

We added to our study *in vivo* recordings of visually-evoked potentials (VEPs) in the primary visual cortex of the awake, free running *rd1* mouse expressing MW-opsin in RGCs. Recordings were made with a 16-channel linear electrode array, with electrodes at 25 micron spacing, allowing us to record single units across the layers of cortex (Fig. S8). Full-field flickering stimuli were presented and we observed that the activity of layer 4 neurons reliably followed frequencies of 0.5, 1, 2, and 4 Hz (Fig. S10).

The heterogeneity across RGCs of *rd1* mouse expressing MW-opsin in RGCs can be seen in the raster plots presented in four of our figures (Figs. 1i, 2a, 2f, and S6d). This heterogeneity can be compared to measurements under similar conditions from a wildtype retina (Fig. S6d versus S6e).

2. Adaptation of MWS responses

The authors show that conditioning with moderate illumination (10 min of 0.1 mW/cm²) causes the sensitivity of MWS responses to shift such that threshold is near 10 mW/cm². Thus, it would appear that the retinas have entirely lost their responses to moderate light intensities by the end

of the conditioning period. This is the case even with a continuous supply of 9-cis retinal. By extension, an MWS animal would have, at most, 10 min of useful vision when going from darkness to a moderately lit room. This is a severe limitation on LWS-mediated vision restoration.

It would be important to examine the responses of MWS-expressing RGCs under continuous illumination at different intensities. To what extent do they decline, and with what time course? Is there a particular range of light intensities over which MWS responses are sustained—that is, where the rate of bleaching/adaptation does not outpace that of regeneration? What is the time course of recovery from adaptation after illuminations of different intensities cease?

We thank the reviewer for bringing up this important point. Of course, if MW-opsin only afforded a few minutes of useful vision when going from darkness to a moderately lit room then this would not be a useful vision restoration therapy. **Fortunately, this is not the case.**

First, the reviewer brought our attention to a typo in the text. The threshold response for retina under light adapted conditions is in fact $100 \mu\text{W}/\text{cm}^2$, not $10 \text{ mW}/\text{cm}^2$. This is displayed in Figure 4b where we observed a shift in sensitivity of about three orders of magnitude after 10 min of $100 \mu\text{W}/\text{cm}^2$ constant light exposure (denoted by the arrow at the top of Fig. 4b). This indicates the excised retina has become insensitive to light below the intensity at which it was adapted, but remains responsive when the intensity reaches or exceeds the adapting intensity. This has been corrected in the text.

To address the question of whether restored light responses would fade away in minutes of exposure to moderate light, we performed new *in vivo* recordings in visual cortex of awake, free running *rd1* mice expressing MW-opsin. We compared responses to moderate light flashes before, during and after 10 mins of background light illumination. We found that responses were maintained (although modestly diminished in size) during the period of background illumination (see below).

But the most important test is visual-guided behavior. And we would want to know that restored vision operates not just after minutes but after hours of being in a lit environment. To test this, we placed MW-opsin expressing *rd1* animals under the same light adapting conditions described

in Fig. 4 (white light; 1 mW cm^{-2} / 535nm spectral component; $50 \text{ } \mu\text{W cm}^{-2}$) for 4 and 8 hours. We then observed their immediate performance in the parallel line image discrimination behavior under the bright image paradigm used in Figs. 3h & 4f. The mice were able to successfully discern between the two patterned objects (Figs. 4f and S13g). The results demonstrate that MW-opsin provides stable vision restoration for many hours under light adapted conditions.

To provide a more generalizable understanding of adaptation, the authors should consider measuring sensitivity as a function of continuous background light of different intensities (not the first and in cones, but Matthews H...Lamb 1990 J Physiol provides a nice illustration of such an experiment). If the kinetics of the flash response changes with background light, that is a sign that adaptation results not just from bleaching but also involves other mechanisms of negative feedback, which would be interesting.

We performed the suggested experiment. We observe that increased background light intensity resulted in little change in either Tau activation or Tau decay of the light response.

Additional Comments

1. The authors state that the sensitivities of cells expressing rhodopsin and MWS cone opsin are similar, but this is not entirely true. Their sensitivity to brief stimulation is similar, but not to steady stimulation—with prolonged illumination, the long duration of the rhodopsin response will allow for temporal summation and thus a higher effective sensitivity than the MWS response. It would be helpful to be clearer on this point.

Clarification has been added to the Results.

2. The MWS response clearly has more than one decay phase; some quantification of the slow phase, and a discussion of how it impacts temporal resolution, is warranted.

Quantification of the slow phase has now been provided in Fig. S4. It appears that the slow phase is more robust in retina following dark adaptation compared to light adaptation (Fig. S4d and Fig. S12a,b). Our new *in vivo* recordings from the visual cortex (layer 4 cells in V1) under light-adapted conditions do not show this slow phase (Fig. S8), and so we do not see a summation during trains of repeated light pulses at up to 4 Hz (Fig. S10).

3. Fig. 2f and g. Does the same relation hold when delivering pulses from shortest to longest duration; that is, in reverse order? Alternatively, is there some assurance that the responses are recovering completely between pulses? From the data shown, each response appears to return to baseline, but it is unclear if sensitivity does.

The responses recover completely between the pulses when the intensity of the pulses is low or the pulses are short, and less completely with long, high intensity pulses, as shown below in an exaggerated form where we compare the amplitude of the responses to the first and fifth pulse in a train using pulses of 10 sec duration. With the short <500ms pulses used in Fig. 2f and g at an intensity of $0.4 \mu\text{W}/\text{cm}^2$ recovery is complete.

4. How consistent is the extent of transduction across animals and retinas?

To determine transfection rate, MW-opsin-YFP expressing retina were stained with 4,6-diamidino-2-phenylindole (DAPI) to stain DNA, as commonly used for automated cell counting in fixed tissue. Z-stacks of the RGC layer across multiple locations of the retina (2 retinas, 10 locations both centrally and peripherally) were imaged in confocal microscopy. The percentage

of cells transfected was determined by co-localization of YFP and DAPI (Imaris software). We determined a $45\% \pm 19\%$ (SD) transfection rate. This is now reported in the Results (line 144/145).

5. What fraction of RGCs detected on the array are photosensitive following transduction?

This is now quantified in Fig. S3. MW-opsin expressing *rd1* retinas were considered light responsive if >10% of the electrodes displayed light sensitivity, a condition met by ~60-70% of retinas from MW-opsin expressing mice. In these responsive retinas, spiking units were detected in 45-58 of the 60 electrodes. Units were classified as light responsive if they had a Light Response Index (LRI) > 0.1. On average, 79% of channels that detect RGCs met this criterion (Fig. S3a). The robustness of the light response was determined by calculating the normalized LRI across cells and retinas (Fig. S3b) and the variability of responses within retina was quantified by correlating all cell responses across each other (Fig. S3c,d). These values were compared to those obtained in wildtype retinas.

6. Why are the intensity-response curves different between Fig. 2b and 4b (the dark-adapted curve)? The latter saturates at a much lower intensity and might also be right-shifted.

Fig. 4b represents a response from cells of a single dark-adapted retina, whereas Fig. 2b is an average of responses from 3 retinas that had some (not equal) prior exposure to light, and so Fig. 2b is expected to be right-shifted and shallower than Fig. 4b, as observed.

7. Throughout the paper, it appears that stimuli of different wavelengths are matched by energy and not photon number, even though it is the latter that is important for photoreception. There should be some discussion of whether this affects the comparisons being made. I would encourage the authors to use photon number in future studies.

We thank the reviewer for their recommendation and will plan to acquire the equipment to perform these measurements for future studies. For the comparisons being made it is known that conversions from energy to photons can be inaccurate. However, our manuscript discusses light intensity broadly, and over various relative magnitudes. Any comparisons are made at a scale so large that differences in units of measurement and the error associated with necessary conversions are unlikely to make a substantial difference.

8. How were comparisons among various opsins made (e.g., the Chr2 minimum in Fig. 4c, Fig. S2)? Is the threshold for each measured at its maximum wavelength of absorption? Were the values reported in the various studies for bandpass filters of similar width, using stimuli that were of comparable duration (because longer stimuli afford more opportunity for temporal integration)? When the experiments were done in vivo, were the pupils dilated, and were expression levels comparable across studies? These comparisons are important to make, but they are complex and should be discussed in a transparent and nuanced fashion.

Thresholds obtained from the literature represent the lowest intensity in which light responses were reported. All studies used electro-physiological recordings (patch clamp or MEA) on excised retina preparations of an *rd* mouse model using AAV as a mode of transfection. Virus

titers were variable (10^{11} - 10^{14}) but expression levels appear comparable. Values were reported using monochromatic wavelengths of light at or near peak excitation. However, the reviewer is correct, there are a number of inconsistencies between the studies that make this comparison challenging. i) The range of band pass filters is broad and in some cases, unreported. ii) Among the studies, different promoters and capsid variants were used for viral transfection, which may display different expression patterns and tissue penetration. iii) The studies differ in what qualified as a response threshold as well as differences in the range of intensity steps implemented by the authors. iv) Intensity was measured using different methods, and expressed in different units (lux, photons, Watts). v) The retinas were isolated from animals at different age, and different strains of *rd* mouse were used. Nevertheless, we have made our best effort to compare the methods to provide a broad sense of the differences in sensitivity between the various approaches.

9. On a related note, it would be helpful to add melanopsin to Fig. S2 if possible.

Melanopsin has been added.

10. In the discussion of how MWS opsin in RGCs would be regenerated in vivo (lines ~350) the authors might consider citing the literature on the visual cycle in Muller glia.

A sentence (and reference; Kaylor et al, 2013) has been added regarding Müller glia as an alternative source of retinal recycling for MW-opsin that is more proximal to the RGCs.

11. Line 299: “MW opsin is fast enough to respond to a flash of light as brief as 25 ms.” This does not quite make sense. The key variable here is whether there are enough photons delivered that enough are absorbed to produce a detectable response. The duration of delivery is practically immaterial--visual pigments activate on the femtosecond time scale.

We have modified this to “at this moderate light intensity, MW-opsin responds to a flash of light as brief as 25ms.”

12. Line 328, it would be helpful to express acuity in terms of cycles per degree.

We have done this.

13. The manuscript should be proofread for typos (e.g., “Coulbourn”) and grammatical errors.

We have done this.

REVIEWERS' COMMENTS:

Reviewer #1 (Remarks to the Author):

The authors have satisfactorily addressed my previous comments. Congratulations on a very nice study.

Reviewer #2 (Remarks to the Author):

The authors' responses to the critiques have further improved an already impressive and important manuscript. The experiments are clear, carefully described and appropriately analyzed to support the claims.